# Membrane binding controls ordered self-assembly of animal septins

**Agata Szuba[1†], Fouzia Bano[2,3,4‡], Gerard Castro-Linares[1,5], Francois Iv[6], Manos Mavrakis[6]\*, Ralf P Richter[2,3,4]\*, Aurélie Bertin[7,8]\*, Gijsje H Koenderink[1,5]\***

[1]AMOLF, Department of Living Matter, Biological Soft Matter group, Amsterdam, Netherlands; [2]School of Biomedical Sciences, Faculty of Biological Sciences, Astbury Centre for Structural Molecular Biology, University of Leeds, Leeds, United Kingdom; [3]School of Physics and Astronomy, Faculty of Engineering and Physical Sciences, University of Leeds, Leeds, United Kingdom; [4]Bragg Centre for Materials Research, University of Leeds, Leeds, United Kingdom; [5]Department of Bionanoscience, Kavli Institute of Nanoscience Delft, Delft University of Technology, Delft, Netherlands; [6]Institut Fresnel, CNRS, Aix-Marseille Univ, Centrale Marseille, Marseille, France; [7]Laboratoire Physico Chimie Curie, Institut Curie, PSL Research University, Paris, France; [8]Sorbonne Université, Paris, France

**\*For correspondence:**
manos.mavrakis@univ-amu.fr (MM);
R.Richter@leeds.ac.uk (RPR);
aurelie.bertin@curie.fr (AB);
g.h.koenderink@tudelft.nl (GHK)

**Present address:** [†]Pollard Institute, School of Electronic & Electrical Engineering, University of Leeds, Leeds, United Kingdom; [‡]Department of Clinical Microbiology, Målpunkt R, NUS, Umeå Universitet, Umeå, Sweden

**Competing interests:** The authors declare that no competing interests exist.

**Abstract** Septins are conserved cytoskeletal proteins that regulate cell cortex mechanics. The mechanisms of their interactions with the plasma membrane remain poorly understood. Here, we show by cell-free reconstitution that binding to flat lipid membranes requires electrostatic interactions of septins with anionic lipids and promotes the ordered self-assembly of fly septins into filamentous meshworks. Transmission electron microscopy reveals that both fly and mammalian septin hexamers form arrays of single and paired filaments. Atomic force microscopy and quartz crystal microbalance demonstrate that the fly filaments form mechanically rigid, 12- to 18-nm thick, double layers of septins. By contrast, C-terminally truncated septin mutants form 4-nm thin monolayers, indicating that stacking requires the C-terminal coiled coils on DSep2 and Pnut subunits. Our work shows that membrane binding is required for fly septins to form ordered arrays of single and paired filaments and provides new insights into the mechanisms by which septins may regulate cell surface mechanics.

## Introduction

Septins are a conserved family of cytoskeletal proteins (*Nishihama et al., 2011*) capable of forming filamentous scaffolds at the cell cortex that participate in many processes such as cytokinesis, cell-cell adhesion, and phagocytosis (*Kartmann and Roth, 2001*; *Huang et al., 2008*; *Dolat and Spiliotis, 2016*; *Marquardt et al., 2019*). Most of what we currently know about the molecular mechanisms by which septins function comes from studies of the budding yeast cell *S. cerevisae*, where septins form hetero-octamers (*Frazier et al., 1998*; *Versele et al., 2004*; *Farkasovsky et al., 2005*; *Bertin et al., 2008*; *Khan et al., 2018*) that form paired filaments (*Byers and Goetsch, 1976*; *Rodal et al., 2005*; *Bertin et al., 2012*). During budding, septins form a collar encircling the bud neck that acts as a scaffold to recruit proteins necessary for cell division (*Longtine et al., 2000*; *Finnigan et al., 2016*; *Tamborrini et al., 2018*), and that restricts lateral diffusion of transmembrane proteins (*Barral et al., 2000*; *Takizawa et al., 2000*; *Clay et al., 2014*). Compared to yeast septins, animal septins have a much more variable cellular localization that changes with tissue type, developmental stage, and cell cycle state (*Bridges and Gladfelter, 2015*; *Spiliotis, 2018*). They are frequently found at curved regions of the plasma membrane such as the base of dendritic spines and

cilia, where they provide a diffusion barrier (*Cho et al., 2011*; *Ewers et al., 2014*; *Palander et al., 2017*). However, animal septins also associate with flat regions of the cell cortex, where they influence the rigidity and contractility of the actin-myosin cortex within single cells (*Tooley et al., 2009*; *Mostowy et al., 2011*; *Gilden et al., 2012*) and within multicellular tissues (*Shindo and Wallingford, 2014*; *Park et al., 2015*; *Kim and Cooper, 2018*). Cortical septins also play a key role in cell division, where they are needed to stabilize the actomyosin ring (*Founounou et al., 2013*) and recruit proteins that mediate chromosome segregation and abscission (*Spiliotis et al., 2005*; *Kim et al., 2011*; *Renshaw et al., 2014*).

Biochemical reconstitution studies have shown that mammalian septins can bind phosphoinositides (*Dolat and Spiliotis, 2016*; *Zhang et al., 1999*; *Tanaka-Takiguchi et al., 2009*; *Bridges et al., 2016*; *Yamada et al., 2016*), but animal septins have also been shown to bind actin filaments (*Mavrakis et al., 2014*; *Dolat et al., 2014*; *Smith et al., 2015*) and various actin-binding proteins including anillin (*Kinoshita et al., 2002*) and non-muscle myosin-2 (*Joo et al., 2007*). Electron microscopy of immuno-stained cells revealed localization of cortical septins with cortical actin in tissue culture cells (*Hagiwara et al., 2011*; *Kinoshita et al., 1997*), but the high density of the actin cortex in animal cells has made it impossible to determine whether cortical septins directly interact with the plasma membrane. It is even unclear whether cortical septins truly form filaments. Fluorescence microscopy has shown the presence of cortical septin puncta, fibrous strands, or rings (*Tooley et al., 2009*; *Mostowy et al., 2011*; *Gilden et al., 2012*; *Park et al., 2015*), but has lacked the resolution to resolve the precise nature of these structures. Septin-deficient cells exhibit a markedly reduced cortical rigidity (*Tooley et al., 2009*; *Mostowy et al., 2011*; *Gilden et al., 2012*; *Park et al., 2015*), but it is unclear whether these effects imply a loss of membrane-bound septin filaments or a loss of septin-mediated actin stabilization. A further complication is that microtubules have also been found to interact with cortical septins (*Sellin et al., 2011*).

Unlike in budding yeast, animal septins associate extensively with the actin and microtubule cytoskeleton, and provide essential functions beyond cell division. Septin removal in nondividing cells and tissues leads to dramatic phenotypes consistent with a loss of cortical integrity, notably the collapse of early gastrulating embryos (*Adam et al., 2000*), loss of sperm integrity (*Kissel et al., 2005*; *Ihara et al., 2005*; *Kuo et al., 2015*), and defects in neuron morphogenesis (*Tada et al., 2007*; *Xie et al., 2007*). Despite the importance of animal septins in these processes, it remains unclear whether the cortical septin pool in animal cells associates with the plasma membrane itself or with membrane-bound actin filaments and microtubules (*Hagiwara et al., 2011*; *Gilden and Krummel, 2010*). To resolve the functions of septins at the animal cell cortex, it is necessary to understand the innate ability of septins to assemble alone and in association with the membrane. Several reconstitution studies have been reported for native and recombinant septins from various animal species, but nearly all these studies considered septin assembly in bulk solution. Frog (*Xenopus laevis*) septins were found to form paired filaments similar to yeast septins (*Mendoza et al., 2002*), while recombinant nematode (*C. Elegans*) septins (*John et al., 2007*) and native and recombinant fly (*Drosophila melanogaster*) septins (*Mavrakis et al., 2014*; *Field et al., 1996*; *Huijbregts et al., 2009*) were observed to form bundles. Recombinant mammalian septin hexamers have been shown to form rings and spirals that are reminiscent of structures formed by Shs1-capped yeast septin octamers (*Garcia et al., 2011*). To the best of our knowledge, there are only two reports until now of the organization of animal septins on lipid membranes, showing that septin-enriched fractions from porcine brain extracts can tubulate giant liposomes and form filamentous meshworks encasing the membrane tubes (*Tanaka-Takiguchi et al., 2009*; *Yamada et al., 2016*).

Here, we investigate the role of membrane-binding in animal septin assembly by reconstituting recombinant animal septins on flat supported lipid bilayers. We focus on septin hexamers composed of Pnut, DSep2, and DSep1 from the model organism *Drosophila*. We studied the septins on model cell membranes composed of phosphatidylcholine (PC) lipids combined with either phosphatidylserine (PS) or phosphatidylinositol(4,5)-bisphosphate (hereafter referred to as PI(4,5)P$_2$). PS is the most abundant anionic lipid in the inner leaflet of the plasma membrane (*Leventis and Grinstein, 2010*). The cell membrane in the fly has been reported to contain 3–10% PS, but the composition varies among tissues and developmental stages (*Carvalho et al., 2012*; *Guan et al., 2013*; *Jones et al., 1992*). PI(4,5)P$_2$ is an anionic signaling lipid previously shown to interact with yeast and mammalian septins (*Zhang et al., 1999*; *Bertin et al., 2010*; *Beber, 2018*) and assumed to be important for septin-membrane interactions in the fly (*Brill et al., 2016*; *Goldbach et al., 2010*). We show by

fluorescence imaging that fly septin hexamers have a high affinity for membranes containing PS and/or PI(4,5)P$_2$, and form dense membrane-associated meshworks. Nanometer-resolution imaging by electron microscopy and by atomic force microscopy reveals that these meshworks comprise single and paired filaments, which laterally associate into bundles that form tightly packed domains. We propose a molecular model of the septin self-assembly mechanism that is consistent with these new data. Our findings establish that membrane binding catalyzes animal septin polymerization and has a dramatic impact on septin self-assembly, with C-terminal coiled-coils playing a key role in higher order septin filament organization.

## Results

### Septin hexamers form bundles in solution

To understand how membranes influence septin assembly, we begin by analyzing how septin oligomers assemble in free solution. We focus on septin hexamers composed of DSep1, DSep2, and Pnut from the model organism *Drosophila*, which have been previously characterized in vivo (*Mavrakis et al., 2014*; *Adam et al., 2000*; *Neufeld and Rubin, 1994*; *Fares et al., 1995*) and in vitro (*Mavrakis et al., 2014*; *Field et al., 1996*; *Huijbregts et al., 2009*), and which are highly homologous to their human septin orthologs (DSep1, DSep2, and Pnut are 75%, 68%, and 65% identical to SEPT2, SEPT6, and SEPT7, respectively; see Materials and methods section). Based on phylogenetic analysis that classifies fly and human septins in distinct subgroups (DSep1/SEPT2 in the SEPT2 subgroup; DSep2/SEPT6 in the SEPT6 subgroup; Pnut/SEPT7 in the SEPT7 subgroup *Cao et al., 2007*; *Pan et al., 2007*), and the recently demonstrated order of subunits in mammalian septin complexes (*Mendonça et al., 2019*; *Soroor et al., 2021*; *DeRose et al., 2020*), fly septin hexamers should have a palindromic arrangement with the Pnut subunits in the center, the DSep1 subunits at the termini, and the DSep2 subunits in between (*Figure 1A*). Each subunit carries a flexible carboxy-terminal extension, a part of which is predicted to adopt a coiled-coil arrangement (*Marques, 2012*). Using the coiled-coil prediction algorithm COILS (see Materials and methods section), we predict the C-terminus of DSep1 to form a coiled-coil that is 28 residues long, and the C-termini of Pnut and DSep2 to form coiled-coils that are 86 residues long (*Figure 1B*). Our coiled-coil prediction analysis for human septins yields the same predictions, that is, 28-residue-long coiled-coils for SEPT2 and 86-residue-long coiled-coils for each SEPT6 and SEPT7 (*Figure 1C*). Considering a 1.5 Å rise per residue in an alpha-helix, we estimate the coiled-coil of DSep1/SEPT2 to be 4 nm long, and the coiled-coils of DSep2/SEPT6 and Pnut/SEPT7 to be 13 nm long (*Figure 1A*). Hexamers form 24-nm-long rods, with the globular domains (G-domains) of the septin subunits approximately 4 nm in diameter (*Mavrakis et al., 2014*; *Sirajuddin et al., 2007*). The coiled-coil of DSep1 could thus extend as much as the G-domain itself, while the coiled-coils of DSep2 and Pnut could even extend as much as threefold the size of the G-domain. From the end of the α6-helix to the start of the predicted coiled-coils, there are stretches of 24, 15, and 15 residues for DSep1, DSep2, and Pnut, respectively (see *Figure 1B*), that are predicted to be unstructured. Given a contour length per residue of ~0.4 nm, these lengths translate to contour lengths of 6–10 nm. These regions thus likely act as a flexible hinge between the G-domain and the coiled-coil, allowing the coiled-coils to pivot around their anchor points. This prediction is consistent with observations that coiled coils did not show up in X-ray crystal structures or particle-averaged electron microscopy (EM) analysis of yeast and mammalian septin oligomers (*Bertin et al., 2008*; *Sirajuddin et al., 2007*). Moreover, EM images of yeast, *Drosophila* and *C. elegans* septin oligomers showed direct evidence that the coiled coils can sweep out a large volume (*Bertin et al., 2008*; *Mavrakis et al., 2014*; *John et al., 2007*).

To test the ability of the recombinant fly septin hexamers to polymerize in bulk solution, we performed TIRF imaging of mEGFP-tagged septin hexamers (*Figure 1D*) after rapid dilution from a high-salt storage buffer containing 300 mM KCl to a low-salt polymerization buffer containing 50 mM KCl. We expect fly septin hexamers to form bundles under these conditions (*Mavrakis et al., 2014*; *Mavrakis et al., 2016*). To enable observation of septins in the thin (100 nm) evanescent TIRF field, we pushed them down onto a coverslip passivated with a neutral (PC) lipid bilayer with the crowding agent methylcellulose at a concentration of 0.1-wt%, which is high enough to crowd the bundles to the surface yet low enough not to cause bundling. As shown in *Figure 1D*, septin hexamers did not form any structures visible at the resolution of the light microscope until the

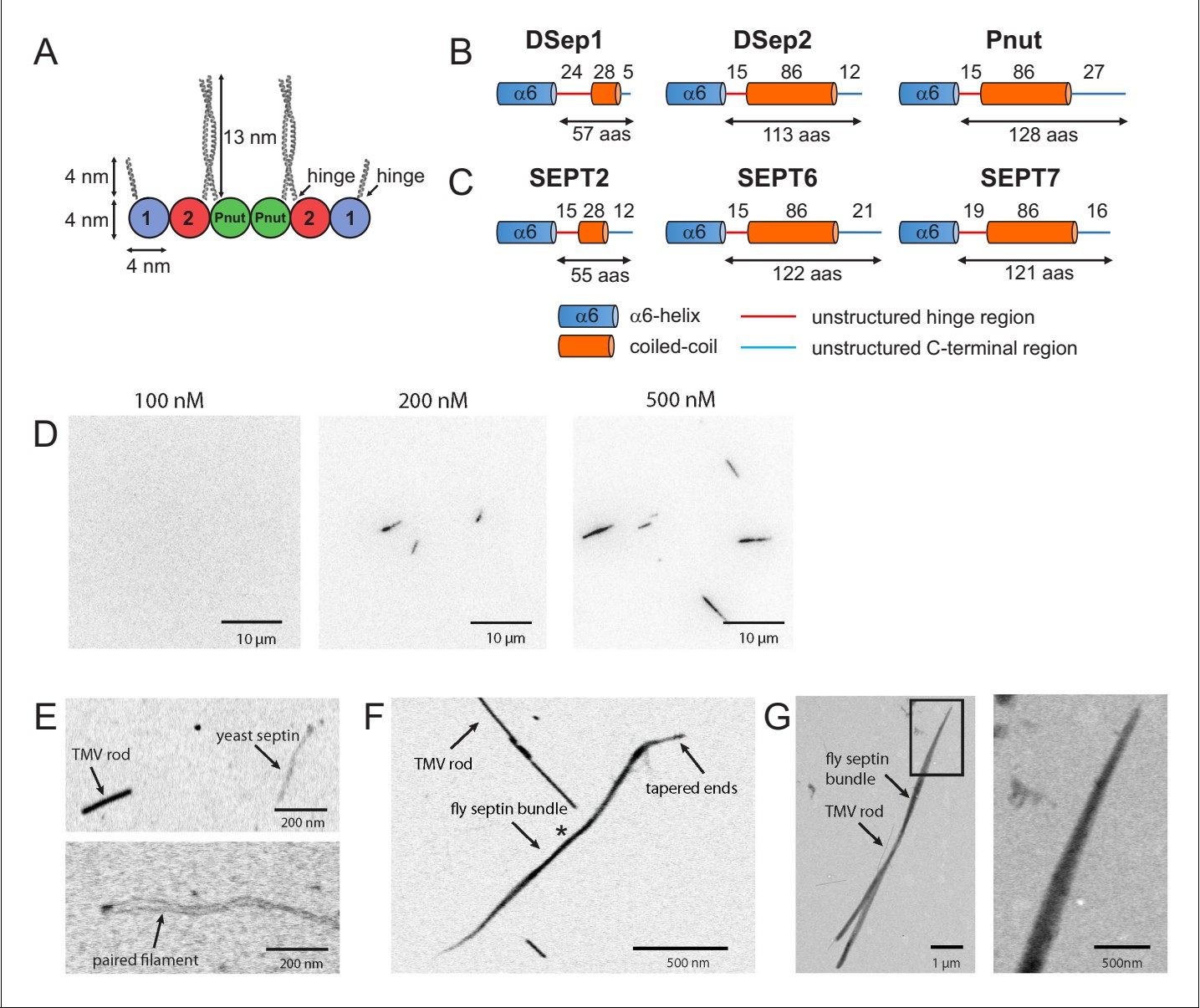

**Figure 1.** Fly septin hexamers form bundles in solution. (**A**) Schematic of the fly septin hexamer, showing its expected palindromic arrangement with long (13 nm) coiled-coil extensions of the Pnut and DSep2 (labelled 2) subunits, and shorter (4 nm) extensions on the DSep1 (labelled 1) subunits. The predicted septin coiled-coils are depicted to scale using available crystal-structures. (**B**) Structure predictions for the C-terminal regions of fly septins, starting from the end of the α6-helix regions (aas – amino acids). (**C**) Corresponding predictions for human septins. (**D**) TIRF images of mEGFP-tagged fly septin bundles formed in solution and crowded onto net-neutral SLBs composed of PC by methylcellulose, with septin concentrations as indicated above each image. (**E**) STEM images of yeast septin filaments. Upper panel: septin filament and a TMV rod (see arrows). Lower panel: paired septin filament. (**F**) STEM image of fly septin bundle formed at a concentration of 200 nM surrounded by several TMV rods (one example is pointed out). The bundle center width and MPL (asterisk) are 31 nm and 250 kDa/nm, respectively. (**G**) STEM image of a fly septin bundle formed at a concentration of 500 nM, together with three TMV rods (one is indicated by an arrow). Note that TIRF and STEM images are all contrast-inverted, so septins appear dark against a light background.

The online version of this article includes the following figure supplement(s) for figure 1:

**Figure supplement 1.** Biochemical and morphological characterization of septin hexamers.

**Figure supplement 2.** Quantification of septin filament MPL from STEM images.

concentration reached 200 nM. When we increased the concentration further to 500 nM, the septin hexamers formed longer bundles. Bundle formation was very rapid, taking less than 2 min for

completion. The bundles were rather straight and did not exhibit measurable thermal bending undulations, indicating that they must be stiff.

To measure the width and mass of the septin bundles, we turned to scanning transmission electron microscopy, which provides quantitative mass measurements based on the dark-field signal from elastically scattered electrons emitted under a high angle (*Sousa and Leapman, 2012*). The sample is raster-scanned with a focused electron beam and the intensity in each image pixel is converted to the projected specimen mass by including tobacco mosaic virus (TMV) rods with a well-defined mass-per-length (MPL) of 131 kDa/nm as an internal calibration. To test the accuracy of the mass mapping method, we first imaged budding yeast septins, since these are already known to form paired filaments with a theoretical mass per length of 23.2 kDa/nm (*Bertin et al., 2008*). As shown in *Figure 1E*, the yeast septins form thin semiflexible polymers that are weaker in intensity than the TMV rods, consistent with their smaller mass per length (*Figure 1—figure supplement 2A, B*). The average mass-per-length (based on 10 septin filaments from three images) was ~20 kDa/nm, close to the mass-per-length expected for paired filaments. In rare cases, the image clearly showed a double-stranded structure with two filaments running in parallel with a small spacing in the range of 17–27 nm, consistent with prior transmission EM findings (*Bertin et al., 2008*) (see lower panel of *Figure 1E*). By contrast, fly septins formed thick bundles that were stronger in intensity than the TMV rods, indicating a larger MPL (*Figure 1F* and *Figure 1—figure supplement 2C*). Bundles formed at 200 nM had tapered ends and a thicker center. The example bundle in *Figure 1F* has a MPL of around 250 kDa/nm in the center (marked by an asterisk) and a corresponding center width of 31 nm. Given a calculated MPL of 12.8 kDa/nm per fly septin hexamer, this number translates to around 20 hexamers per cross-section. Considering that the inter-filament spacing has to be at least 4 nm, a width of 31 nm and 20 hexamers per cross-section implies there are at least three monolayers. At 500 nM, the septin bundles were thicker (*Figure 1G*) with maximal widths of up to 280 nm and MPL values up to ~5000 kDa/nm (*Figure 1—figure supplement 2D*), corresponding to >400 hexamers per cross-section or a thickness of at least six monolayers. This analysis suggests that septin bundles in solution do not grow as flat sheets but as 3D bundles.

## Lipid membranes recruit septins and promote their assembly

To investigate how membrane-binding affects septin hexamer polymerization, we deposited the purified fly septin hexamers in polymerization buffer on glass-supported lipid bilayers (SLBs) composed of net-neutral PC lipids doped with anionic lipids (*Figure 2A*). To test whether fly septin hexamers bind $PI(4,5)P_2$, as reported for yeast septins (*Bertin et al., 2010*; *Beber, 2018*), we doped the PC membranes with different mole percentages of $PI(4,5)P_2$ ranging from 1% to 8%. Already at 1% $PI(4,5)P_2$, septin hexamers formed immobile bundles adhering to the membrane (*Figure 2B*). Importantly, membrane recruitment was observed here in the *absence* of methylcellulose, showing that septin hexamers indeed bind to $PI(4,5)P_2$ lipids. As we raised the $PI(4,5)P_2$ content from 1% to 8%, we observed a striking transition from a sparse arrangement of thick bundles to a dense protein layer. At intermediate $PI(4,5)P_2$ fractions of 2.5% and 5% we could still discern septin bundles, but at 8% $PI(4,5)P_2$ the protein density was too high to discern any details of the ultrastructure.

To test whether fly septin hexamers selectively bind the $PI(4,5)P_2$ head group as reported for yeast septins (*Bertin et al., 2010*; *Beber, 2018*) or simply bind through nonspecific electrostatic interactions, we next replaced $PI(4,5)P_2$ by PS. As shown in *Figure 2C*, fly septin hexamers were also recruited to the PS-containing bilayers and again showed a transition from a sparse distribution of thick bundles at low PS content to a dense layer at high PS content. These experiments were again carried out in the absence of methylcellulose (except for the reference image with 0% PS, where methylcellulose was required to push septin bundles down into the TIRF field). The filamentous septin structures were already present as soon as we could start imaging (~3 min after septin loading into the flow channel) and the structures did not change in number or thickness over time (*Figure 2— figure supplement 1A*). Apparently, septin hexamers bind strongly and rapidly to the membrane. The total fluorescence intensity, which is a proxy for the amount of membrane-bound septins, linearly increased as the PS content was raised from 5% to 20% (*Figure 2—figure supplement 1B*). The observation that PS has a qualitatively similar effect compared to $PI(4,5)P_2$ on septin recruitment and assembly on SLBs suggests that the membrane-binding affinity of fly septin hexamers is governed by the net surface charge of the membrane rather than by any specific affinity for $PI(4,5)P_2$. The transition from dilute septin bundles to dense septin films required a larger PS mole fraction (10%) than PI

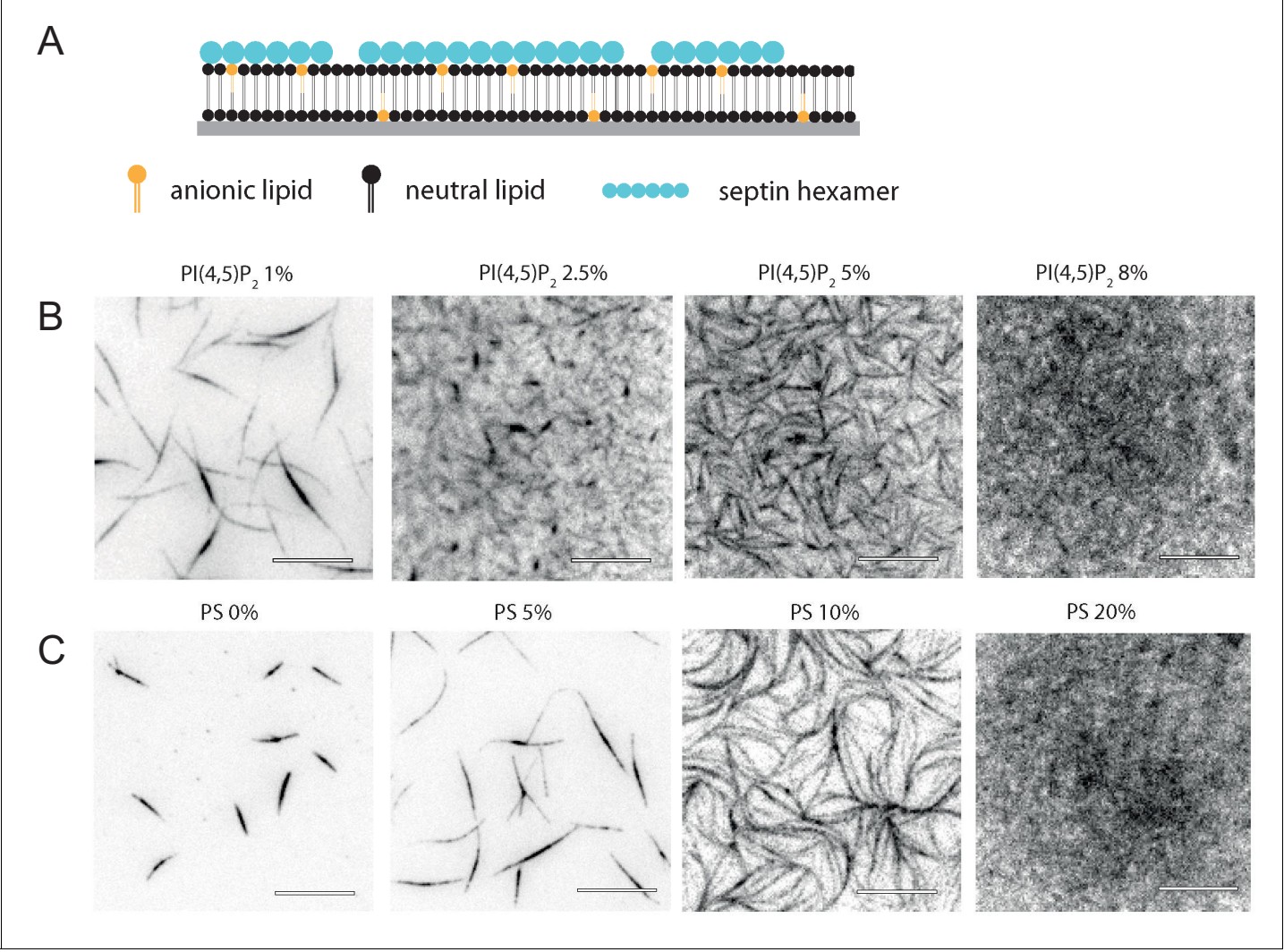

**Figure 2.** Glass-supported lipid bilayers containing anionic lipids recruit septins and promote the assembly of dense septin meshworks. (**A**) Purified fly septin hexamers are deposited on a glass-supported bilayer composed of net-neutral (PC) and anionic (PS or PI(4,5)P$_2$) lipids (sketch not to scale). (**B**) TIRF images recorded ~3 min after the deposition of septin hexamers (1 μM; 10 mol-% mEGFP-tagged hexamers) on bilayers doped with PI(4,5)P$_2$ at mole fractions between 1% and 8% (see legend). (**C**) TIRF images obtained ~3 min after the deposition of septins (1 μM) on bilayers doped with PS at mole fractions between 0% and 20% (see legend). Note that methylcellulose was used to crowd septin bundles to the bilayer for the neutral SLB (0% PS) but was left out in all other cases. All images are contrast-inverted, so septins are dark and membrane areas devoid of septins are light. Scale bars: 10 μm.

The online version of this article includes the following figure supplement(s) for figure 2:

**Figure supplement 1.** Dependence of septin adsorption on septin concentration and on the PS content of the SLBs.

**Figure supplement 2.** FRAP data testing the mobility of septins and lipids.

(4,5)P$_2$ (2.5%), consistent with the larger net charge (−4) of the head group of PI(4,5)P$_2$ due to the phosphate groups compared to the net charge (−1) on the head group of PS (*Blin et al., 2008*; *Kooijman et al., 2009*; *Toner et al., 1988*; *Graber et al., 2012*). Therefore, our findings suggest that the total negative surface charge of the membrane governs septin hexamer adsorption, indicating that the septin-membrane interaction is primarily electrostatic in origin.

To screen for the dependence of septin hexamer adsorption on bulk septin concentration, we incubated bilayers containing 20 mole-% PS with solutions of septin hexamers over a wide range of concentrations spanning from 10 to 300 nM (*Figure 2—figure supplement 1C*). To obtain a bright enough signal, we performed these experiments with 100% GFP-tagged hexamers. At low concentrations (10–50 nM), we observed dim puncta and also brighter puncta, indicative of protein

aggregation (note that images are contrast-inverted). We could not distinguish clear evidence of filaments. We suspect that the bright puncta are due to GFP-mediated aggregation because we observed fewer bright puncta when we co-polymerized 10% GFP-tagged hexamers with 90% dark hexamers. Remarkably, when we increased the septin concentration to values of 100 nM and higher, we suddenly observed a fibrillar meshwork of septin filaments. At concentrations above 200 nM, we found by off-TIRF-angle imaging that, in addition to the membrane-bound septin meshwork, also septin bundles were present in the solution above the bilayer, consistent with the observed onset for bundling in the absence of an adhesive membrane (*Figure 1D*). Fluorescence recovery after photobleaching experiments showed that the membrane-bound septins exhibited negligible subunit exchange (*Figure 2—figure supplement 2A*), indicating that they are stably anchored to the membrane. The membrane underneath the septin film was nevertheless fluid, as indicated by FRAP measurements of rhodamine-labeled PE tracer lipids (*Figure 2—figure supplement 2B*).

To discern the ultrastructure of membrane-adsorbed septins, we turned to transmission electron microscopy. We incubated septin hexamers with lipid monolayers, deposited these on EM grids, and negatively stained the protein with 2% uranyl formate, following a protocol previously used with yeast septins (*Bertin et al., 2010*). To test the role of membrane composition, we formed lipid monolayers by mixing PC lipids with either 20 mol-% PS, 5 mol-% PI(4,5)P$_2$, or a combination of both that mimics the co-existence of both lipids in the plasma membrane of cells (*McLaughlin and Murray, 2005*; *Yeung et al., 2008*). We chose a septin hexamer concentration of 65 nM, below the bundling threshold (200 nM), to ensure that septin self-assembly was initiated at the membrane. As shown in *Figure 3A–C*, septin hexamers formed densely packed arrays of thin filaments in all three cases, which is qualitatively consistent with the dense meshworks observed by TIRF microscopy. Close inspection of the EM micrographs revealed paired septin filaments (black arrows point out examples) with an average center-to-center spacing of 5.7 ± 0.8 nm on monolayers containing PS plus PIP(4,5)P$_2$ (see *Figure 3—figure supplement 1A*). The width of the individual filaments was in the range of 3.5–4 nm (3.6 ± 0.53 nm ($N$ = 60) averaged over all lipid conditions; *Figure 3—figure supplement 1B*), consistent with the expected 4 nm width of fly septin hexamers (*Mavrakis et al., 2014*).

On monolayers containing 20% PS plus 5% PI(4,5)P$_2$, we could observe clear examples of paired filaments that were bridged at intervals of 24 nm by single or double hexamer rods (*Figure 3C*), recognizable by their lengths clustering around either 24 nm or 48 nm (*Figure 3—figure supplement 1C*), respectively. These bridges were either perpendicular to the filaments they bridged (blue arrowheads) or under an angle (red arrowheads; see *Figure 3—figure supplement 1C* for quantification). We observed comparable arrays of tightly paired filaments connected by orthogonally or diagonally oriented single filaments for mammalian septins on lipid monolayers of the same composition (*Figure 3—figure supplement 2*). Similar arrays of filaments were previously also observed for budding yeast septins (*Garcia et al., 2011*; *Bertin et al., 2010*), suggesting that this architectural feature is conserved within eukaryotes. We do note a subtle difference for the mammalian septin hexamers as compared to the fly septins. For the fly septin hexamers, the bridges between paired filaments are single or double isolated hexamer rods. Instead, for the mammalian septin hexamers, the paired filaments (white arrows in *Figure 3—figure supplement 2*) are intersected with a distinct set of thinner continuous filaments (red arrows), creating a network of interconnected and perpendicular filaments. It is unclear whether the bridging filaments are single or paired; if paired, the filaments could be rotated or twisted to appear thinner.

Since the sample preparation for the EM experiments requires drying and negative staining, we corroborated the findings from the lipid monolayer assays by cryo-EM imaging on lipid bilayers using large unilamellar vesicles (LUVs) comprised of PC and 5 mol-% PI(4,5)P$_2$ incubated with septin hexamers. We chose a septin hexamer concentration of 160 nM, higher than the 65 nM concentration used in the lipid monolayer experiments, because we expected reduced binding to the positively curved LUV membranes (*Beber et al., 2019*). As shown in *Figure 4*, the cryoEM images confirm the presence of paired septin filaments on PI(4,5)P$_2$-containing lipid membranes (examples are traced out by double red lines in *Figure 4A–B*). The average center-to-center distance between the filaments forming a pair was 5.7 ± 0.8 nm ($N$ = 40, see *Figure 3—figure supplement 1* and *Figure 4—figure supplement 1*), in excellent agreement with the negative stain images of septin filaments on lipid monolayers. In addition, we occasionally observed single filaments or hexamers (examples are traced out by single red lines in *Figure 4B*) including some that appeared to

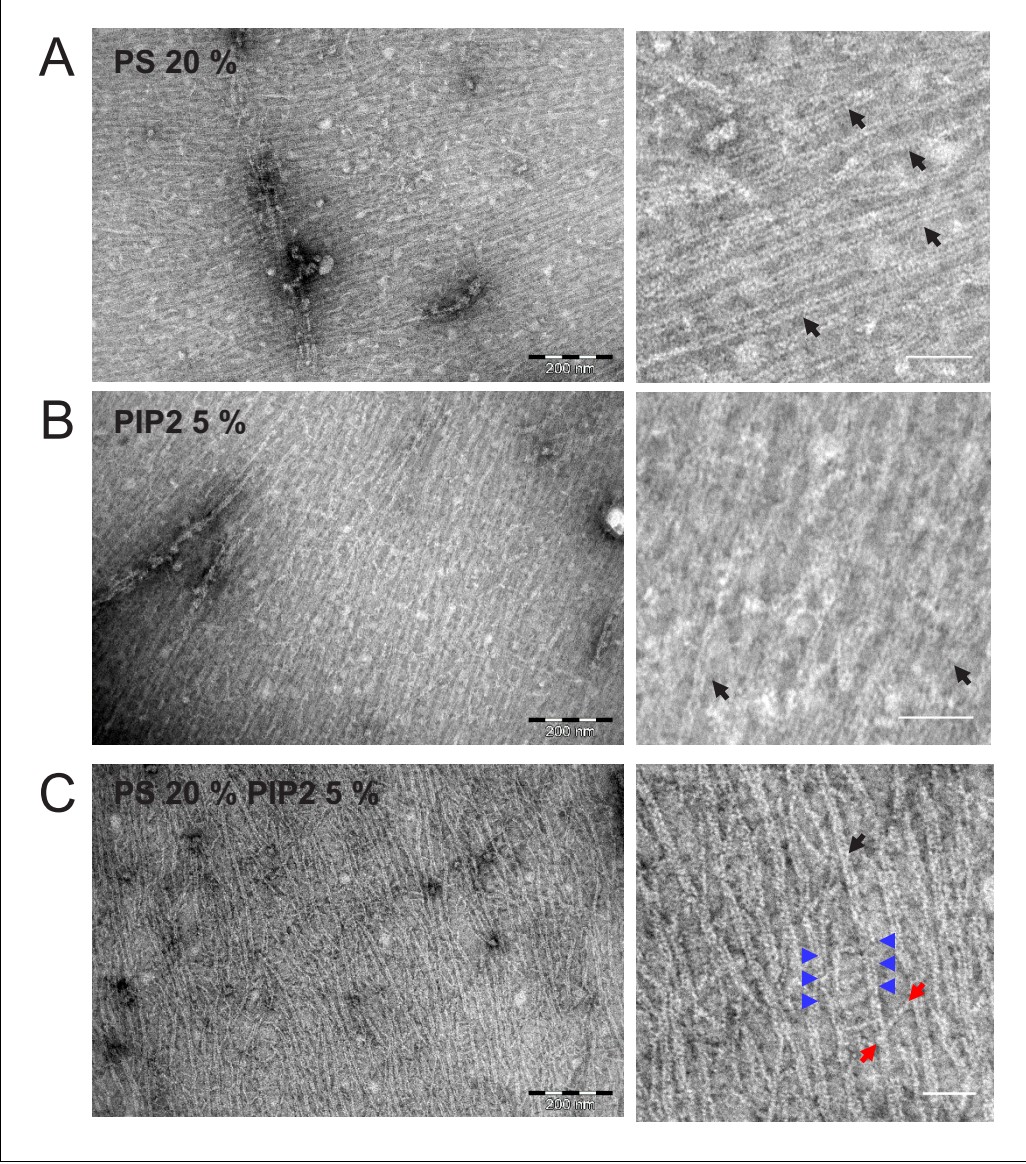

**Figure 3.** Septin hexamers form dense arrays of tightly paired filaments on anionic lipid monolayers. Electron micrographs of negatively stained fly septin hexamers (65 nM) after overnight incubation with lipid monolayers composed of PC combined with anionic lipids: (**A**) 20% PS, (**B**) 5% PI(4,5)P$_2$, and (**C**) 20% PS plus 5% PI(4,5)P$_2$. Images on the right show zoomed-in regions of the images on the left. Black arrows point out examples of paired filaments recognizable by two linear filaments running in parallel. Blue arrows indicate examples of orthogonal hexamers, recognizable by their ~ 24 nm length, between adjacent paired filaments. The two red arrows point to the two ends of a longer (43 nm) cross-bridging filament that bridges two adjacent paired filaments under an oblique angle. Scale bars: 250 nm (left) and 50 nm (right).

The online version of this article includes the following figure supplement(s) for figure 3:

**Figure supplement 1.** Quantification of filament dimensions from electron microscopy images.

**Figure supplement 2.** EM images of mammalian septin hexamers on lipid monolayers.

interconnect filaments at an orthogonal or oblique angle (blue lines in *Figure 4B*). The density of filaments varied among vesicles (see *Figure 4* and *Figure 4—figure supplement 2*), which might be due to some variability in the proportion of charged lipids incorporated within the vesicles. Besides, membrane binding could potentially be curvature-dependent, as was shown for yeast septin octamers reconstituted on membrane-coated beads, rods, and wavy substrates (*Beber et al., 2019*; *Cannon et al., 2019*). An indication for curvature-dependent binding in our data is that the diameter

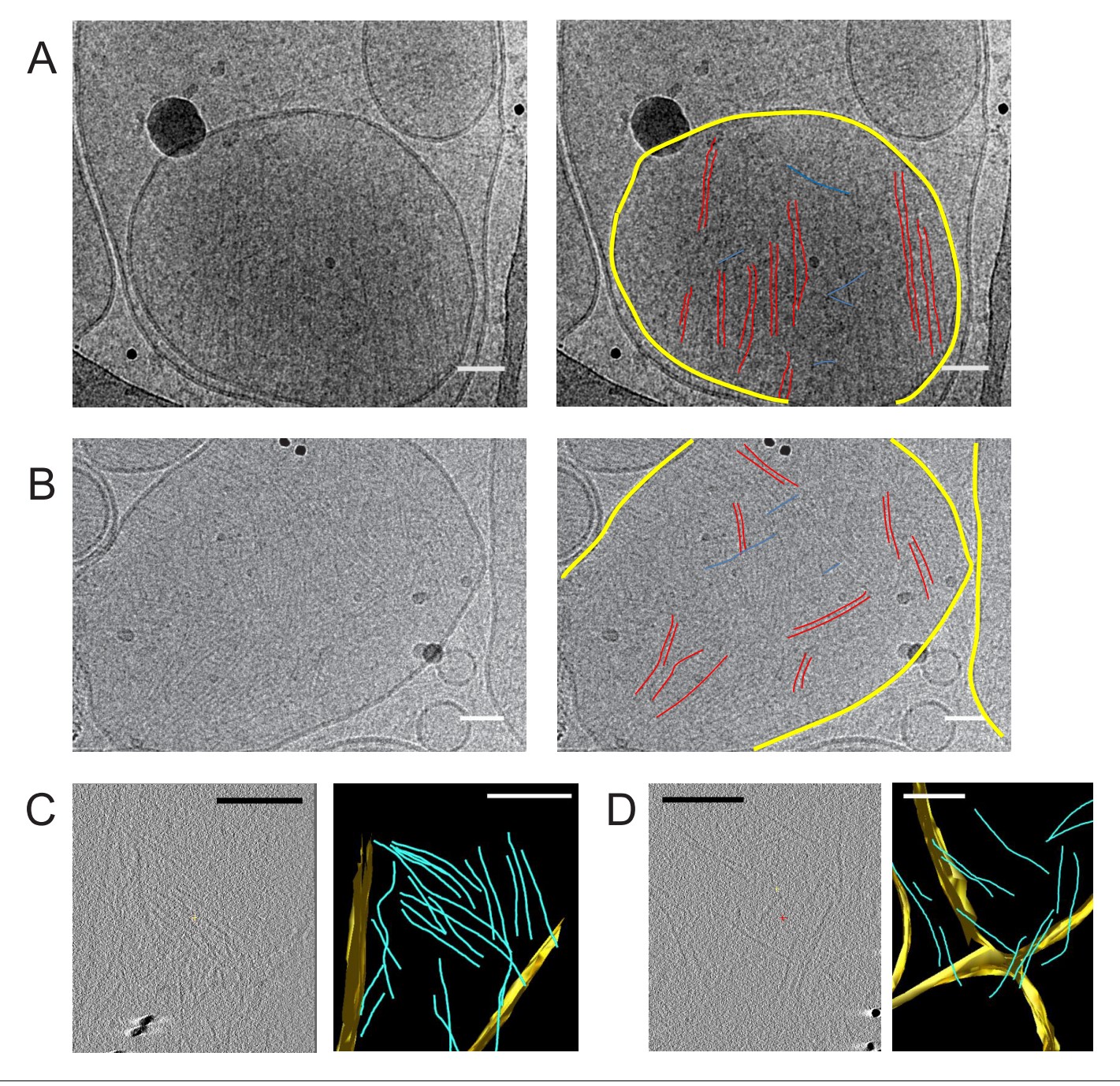

**Figure 4.** Septin hexamers form single and paired filaments on large unilamellar lipid vesicles. CryoEM images of fly septin hexamers (160 nM) after a 30 min incubation with LUVs containing 95% PC and 5% PI(4,5)P$_2$. (**A**) Example image (left) with paired septin filaments traced out in red, connecting orthogonal filaments in blue and the vesicle membrane highlighted in yellow (right). (**B**) Another example image (left), with mostly paired and occasionally single filaments traced out in red, connecting orthogonal filaments in blue and membranes in yellow (right). Black dots are gold nanoparticles that were included as fiducial markers for tomography. The black curved lines in panels A, B come from the carbon lacey substrate. (**C**) Slice from 3D reconstructed cryo-tomogram (left) with segmented data (right). (**D**) Another example tomogram (left) with segmented data (right). Reconstructions show membrane in yellow and septin filaments in blue. Note that the bilayer perpendicular to the electron beam is poorly defined because of the missing wedge. Scale bars are 50 nm in A,B and 850 nm in C, D.

The online version of this article includes the following video and figure supplement(s) for figure 4:

**Figure supplement 1.** Interfilament spacing for paired fly septin filaments.

**Figure supplement 2.** Additional cryoEM data.

*Figure 4 continued on next page*

*Figure 4 continued*

**Figure supplement 3.** Quantification of vesicle size distributions.
**Figure 4—video 1.** 3D tomographic reconstruction of septins bound to lipid vesicles.
https://elifesciences.org/articles/63349#fig4video1

of control vesicles was measured to be 358 ± 180 nm (*N* = 40) while the diameter of similar vesicles displaying septins was 542 ± 199 nm (*N* = 24), thus slightly higher, on average (*Figure 4—figure supplement 3*). This may suggest that septins preferentially interact with liposomes of lower curvatures. Vesicles decorated with septin filaments typically appeared to have a faceted contour, whereas vesicles alone usually retained their spherical shape in cryoEM. The deformed membranes provide a first indication that septins were bound to the membrane and able to flatten it. Some septin filaments clearly protrude a bit beyond the vesicle they adhere to, suggesting they were pre-polymerized in solution and attached to one side of the vesicle without fully wrapping around it. Cryo-electron tomography gave further clear evidence that septin filaments are indeed membrane-bound (see *Figure 4C–D* and *Figure 4—video 1*).

## Septins form thin and mechanically resilient membrane-bound arrays

We further complemented EM imaging with AFM experiments, which allowed us to image septins on flat lipid bilayers under hydrated conditions resembling the conditions in the TIRF experiments. We performed these experiments on silica-supported lipid bilayers containing 20% PS, which have already been extensively characterized in terms of their quality by AFM and other biophysical techniques (*Richter et al., 2003*). In particular, the even inter-leaflet distribution of PS lipids in liposomes is approximately preserved upon SLB formation on silica (and glass) supports, whereas this is not the case on mica (*Richter et al., 2005*), another commonly used support for AFM. We first tested the dependence of septin hexamer assembly on septin concentration by imaging septins at concentrations of 12, 24, and 60 nM (*Figure 5A–B*). We chose this narrow concentration range based on the TIRF data, which showed that septin assembly is restricted to the membrane surface as long as the septin hexamer concentration is below 200 nM. In most experiments, the samples were fixed with 1% glutaraldehyde (GTA) to prevent septin disruption by the AFM tip.

At 12 nM, septin hexamers formed threads of typically several micrometers in length that were sparsely and randomly distributed on the bilayer surface (*Figure 5A–B*, left column). These were mostly isolated but could also be seen to meet, mostly at a shallow angle, and merge. Their apparent height was mostly uniform at 5.1 ± 0.9 nm (*N* = 29), with a few notable exceptions around 12 nm (*N* = 2). The apparent width showed two relatively broad but distinct populations: most isolated threads had a width of 11.5 ± 1.6 nm (*N* = 19), whereas all merged threads and a few isolated ones were wider (16.6 ± 0.9 nm; *N* = 12) (*Figure 5C*, left column, and *Figure 5—figure supplement 1*). These values are consistent with a mixture of mostly single and paired filaments. This can be appreciated if one considers that the minimal filament height is defined by the extension of the globular domain (4 nm) plus some additional contribution by the coiled coils which, owing to the flexible hinge, may point in various directions and additionally become flattened by the force exerted by the AFM tip. Moreover, the flexing of the coiled coils and tip convolution effects (see Materials and methods for details) can explain the relatively broad width distributions, and why the apparent mean widths exceeded the widths of single and paired filaments as seen by EM by about 8 nm on average. The sparse surface coverage observed at 12 nM is consistent with the coverage expected for binding from a semi-infinite and still bulk solution to a planar surface. Mass-transport limitations provide a robust upper limit for the amount of bound protein as $\Gamma \leq 2c\sqrt{Dt/\pi}$ (*Hermens et al., 2004*). With an incubation time *t* = 15 min, a septin concentration *c* = 10 nM, and diffusion constant *D* ≈ 20 μm²/s (a rough estimate for molecules with the size of septin hexamers in aqueous solution), this gives $\Gamma \leq 0.10$ pmol/cm². Further assuming a footprint of 4 nm × 24 nm per septin hexamer, the expected surface coverage is $\leq 6\%$, consistent with the sparse coverage observed by AFM.

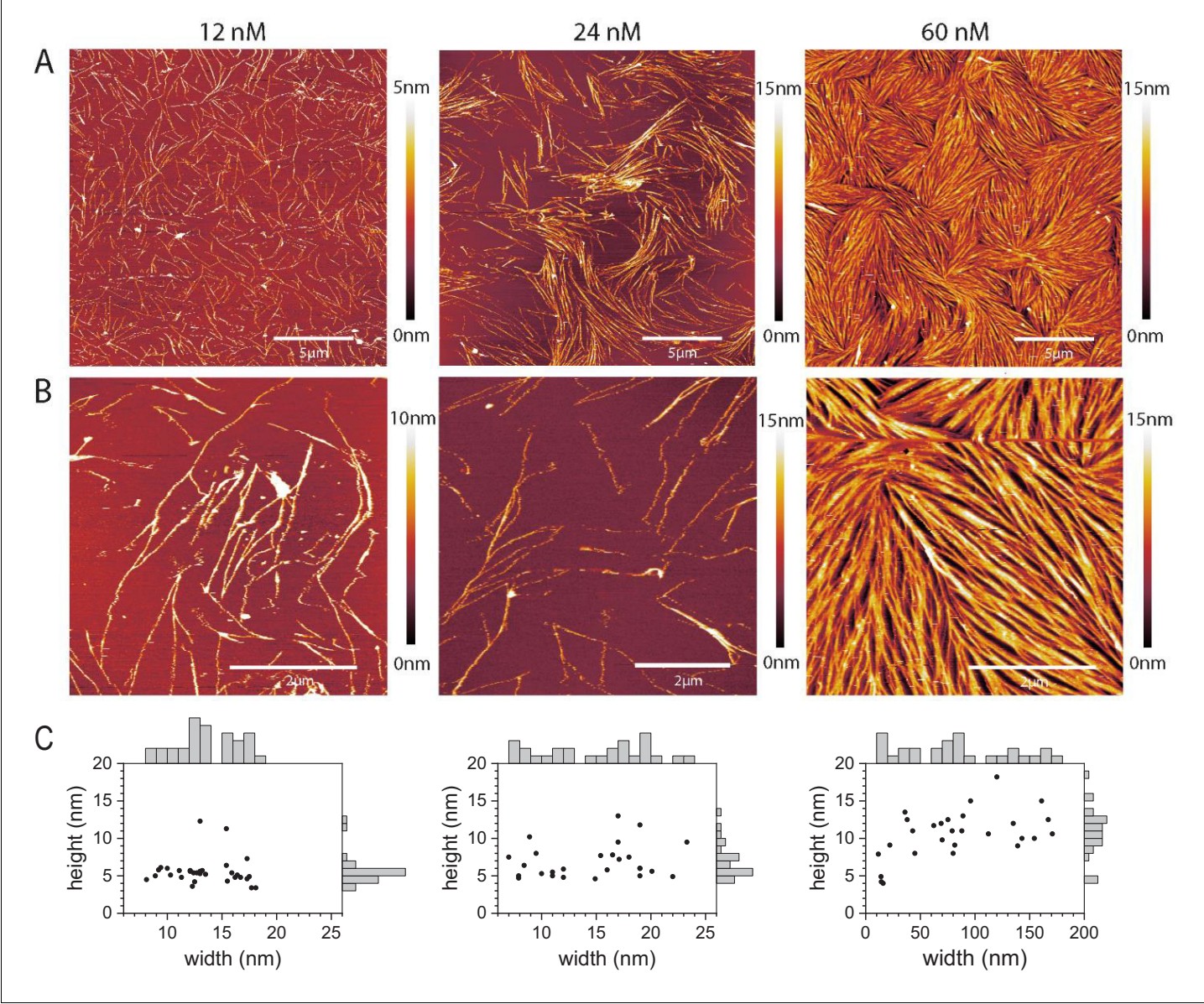

**Figure 5.** Septins form filaments and ordered arrays on lipid bilayers composed of 80% PC and 20% PS. (A–B) AFM topographic images of membrane-bound fly septin structures formed at solution concentrations of 12 nM (left), 24 nM (middle) and 60 nM (right) and observed at a scan size of (A) 20 × 20 μm². (B) Same samples, imaged at a scan size of 5 × 5 μm² (left and right) and of 6.7 × 6.7 μm² (middle). Color bars on the right show the height scale. The samples were fixed with glutaraldehyde (GTA). (C) Scatter plots with marginal histograms of thread widths and heights determined from the corresponding images in (B). N = 31 (left), 26 (middle), and 27 (right) measurements were taken, respectively, across representative sets of well-resolved threads.

The online version of this article includes the following figure supplement(s) for figure 5:

**Figure supplement 1.** Examples of AFM height profiles across filaments and bundles.

**Figure supplement 2.** Images of non-fixed septin samples on lipid bilayers.

**Figure supplement 3.** AFM experiment testing the mechanical stability of the septin ultrastructures.

**Figure supplement 4.** AFM experiment showing septin bundles can be displaced along the membrane.

At 24 nM, septin hexamers formed threads that were isolated in some places and concentrated in areas of enhanced density in other places (*Figure 5A–B*, center column). Thread heights and widths were comparable to the 12 nM conditions, although with a higher proportion of higher (up to 13

nm) and wider threads (*Figure 5C*, center column), suggesting the increased presence of paired filaments and the initiation of bundles made from more than two filaments.

At 60 nM, the septin hexamers formed threads that densely covered the bilayer surface (*Figure 5A–B*, right column) in closely apposed patches of aligned filaments, resembling nematic domains observed for other 2D arrays of densely packed semiflexible biopolymers (*Zhang et al., 2018*). Salient features of the patches were that the constituent threads varied in height and width, and that they formed a network that is characterized by threads frequently joining and disjoining at a fairly shallow angle of approximately 15 degrees. The smallest thread heights and widths observed were comparable to those observed at 12 nM, indicating isolated single or paired filaments were still present. Other threads appeared much wider (up to a few 100 nm; *Figure 5C*, right column). This suggests that many (up to several tens of) single filaments may closely align on the lipid membrane, although the lateral resolution was insufficient to reveal the individual filaments and their spacing. The wider threads also had an elevated height, mostly between 8 and 13 nm and occasionally up to 18 nm (*Figure 5C*, right column), suggesting that septins also stacked on top of each other.

Importantly, the morphologies of unfixed septins resembled those of GTA-fixed ones, though imaging in this case was challenging because the filaments were more easily disrupted by the AFM tip (*Figure 5—figure supplement 2*). For one 60 nM unfixed sample, we came across bilayer areas where the septin coverage was low enough to reveal septin filaments that were isolated or ran close together and in parallel with others (*Figure 6*). Here, we observed that the isolated filaments had heights of 4 nm, corresponding to the height of a single septin hexamer and thus a single layer of septin filaments, while the bundled areas had heights between 8 nm and 12 nm, suggesting that septin filaments can stack on top of each other. We note that in the EM data there is no such clear evidence of layering. However, we note several experimental differences between AFM and EM. In AFM we used solid-supported lipid bilayers as a membrane substrate for septins, whereas in EM we used either lipid monolayers at the air-water interface (for negative stain images) or vesicles (for cryoEM), where the curvature likely prevents septins from reaching a high density of filaments.

Since AFM imaging involves mechanical scanning across the surface, it allowed us to qualitatively test how firmly the septins are attached to the lipid bilayer by performing multiple consecutive scans. In sparsely covered bilayer regions, wider bundles generally remained stable whereas narrower bundles or isolated filaments (both GTA-fixed and unfixed) sometimes appeared ragged, suggesting

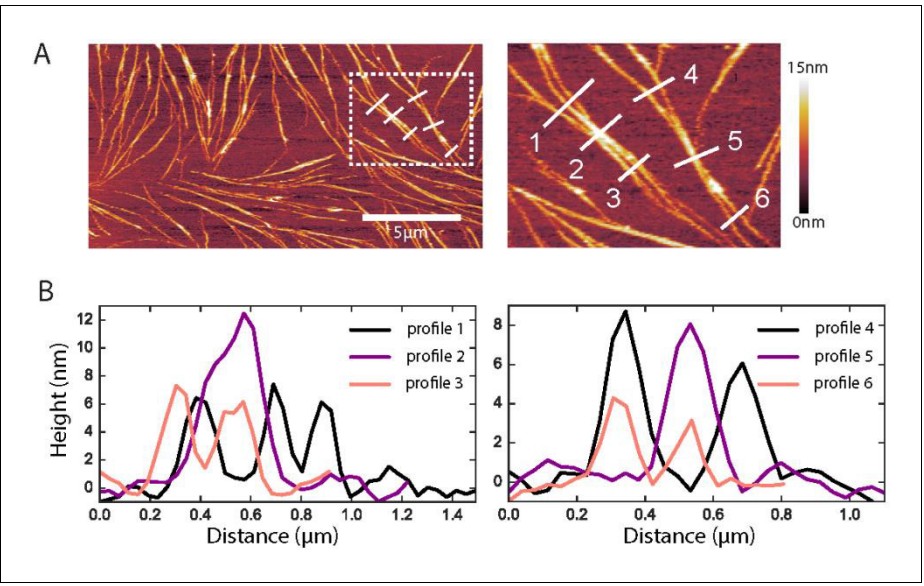

**Figure 6.** Native ultrastructure of septin assemblies on lipid bilayers composed of 80% PC and 20% PS. (**A**) AFM image of non-fixed septins at 60 nM, showing a bilayer region that happens to be sparsely covered with septin filaments. Scale bar for the left panel: 5 µm; right panel shows zoom of the dashed rectangle in the left panel; color bar on the right shows the height scale. (**B**) Height profiles were determined along the numbered white lines in the images in A.

that the AFM tip moved them along the membrane (*Figure 5—figure supplement 3A,B*). Densely covered regions formed at 60 nM septins were completely unchanged after three consecutive 10 min scans, again both for GTA-fixed and unfixed samples (*Figure 5—figure supplement 3C,D*). These observations suggest that lateral associations among septin filaments within bundles or dense arrays may cause mechanical stabilization. Some image sequences clearly showed that entire bundles can be laterally displaced along the membrane owing to lateral forces exerted by the AFM tip. Interestingly, these events did not destroy the bundles but resulted in permanent local kinks, suggesting the septin filaments are intrinsically stiff and ductile but the link to the membrane is fluid. We again observed this behavior both for GTA-fixed and unfixed samples (*Figure 5—figure supplement 4A, B*).

## QCM-D measurements show self-limiting septin assembly away from the membrane plane

The AFM data suggest that fly septin hexamers form organized layers of stiff filaments on PS-containing bilayers, with a limited thickness under conditions where assembly is initiated at the membrane (i.e. at septin concentrations below 200 nM). The maximum thickness of the septin films at 60 nM was approximately 18 nm, according to the height profiles. However, we note that this value may underestimate the actual geometrical height because the tip can potentially indent or otherwise disturb the septin layer. To independently measure the film thickness, and to gain insight into the kinetics of septin film formation, we therefore turned to quartz crystal microbalance with dissipation monitoring (QCM-D), an acoustic technique that measures the mass and variations in the mechanical properties of interfacial films in real time (*Reviakine et al., 2011*). We first formed an SLB on the silica-coated QCM-D sensor by perfusing the channel with a suspension of sonicated unilamellar vesicles (SUVs) and next perfused with a solution of septin hexamers. To investigate the kinetics of SLB formation and septin binding, we monitored the resulting shifts in the resonance frequency $\Delta f$, which is proportional to the areal mass density of adsorbed biomolecules plus hydrodynamically coupled solvent, and in the dissipation $\Delta D$, an indicator of the mechanical properties of the adsorbed layer.

Typical QCM-D data are presented in *Figure 7*. The SUVs are composed of 80% PC and 20% PS, as in the AFM experiments. SUV perfusion on a plain silica surface (*Figure 7A*) caused immediate changes in $\Delta f$ and $\Delta D$ in a two-stage process that is characteristic for the initial adsorption of intact SUVs to the sensor surface followed by SUV rupture, spreading and coalescence (*Richter et al., 2006*). The final shifts of $\Delta f = -25 \pm 1$ Hz and $\Delta D < 0.2 \times 10^{-6}$ ($N = 6$) are typical for confluent SLBs composed of PC and PS of high quality, that is, with a negligible quantity of residual intact vesicles (*Richter et al., 2003*). Subsequent rinsing with vesicle buffer did not result in appreciable changes in $\Delta f$ and $\Delta D$, confirming the SLB was stable. Septin hexamer perfusion at 60 nM on such a SLB (*Figure 7B*) produced an immediate decrease of $\Delta f$ and a concurrent increase of $\Delta D$, indicating that septins adsorbed. Binding reached a plateau after about 50 min, and subsequent perfusion with buffer caused no appreciable change in $\Delta f$, suggesting that septin hexamers stably adsorb to the membrane. Moreover, the saturation of the signal despite continued perfusion with septin hexamers indicates the presence of a mechanism that limits septin binding and film growth. We observed similar binding kinetics and self-limiting levels of binding on SLBs containing 5% PI(4,5)P$_2$, or a combination of 5% PI(4,5)P$_2$ and 20% PS (*Figure 7—figure supplement 1A,B*). In contrast, we did not observe any adhesion of septin hexamers to pure PC membranes (*Figure 7B*; dotted blue line with diamonds). We conclude that the presence of negatively charged lipids is required for septin-membrane binding, consistent with the TIRF observations, and that film growth is self-limiting.

The total frequency shift for septin binding was $\Delta f$ = -93 $\pm$ 4 Hz ($N = 4$) on SLBs containing 20% PS, $\Delta f$ = -77 $\pm$ 2 Hz ($N = 2$) on SLBs containing 5% PI(4,5)P$_2$, and $\Delta f$ = -113 $\pm$ 7 Hz ($N = 2$) on SLBs containing both 5% PI(4,5)P$_2$ and 20% PS (*Figure 7—figure supplement 1C*). Using the Sauerbrey equation (see Eq. (1) *Nishihama et al., 2011*), we estimate septin film thickness values of 15 nm on SLBs containing 20% PS, 13 nm on SLBs containing 5% PI(4,5)P$_2$, and 19 nm on SLBs containing both 5% PI(4,5)P$_2$ and 20% PS, from the respective frequency shifts. This is consistent with the upper end of bundle heights measured by AFM (*Figure 5C*, right column), which we had attributed to stacking into more than one layer.

The dissipation shifts are consistent with the septin film having soft molecular linkers within the structure. To a first approximation, the $\Delta D/-\Delta f$ ratio is a measure of elastic compliance (i.e. softness)

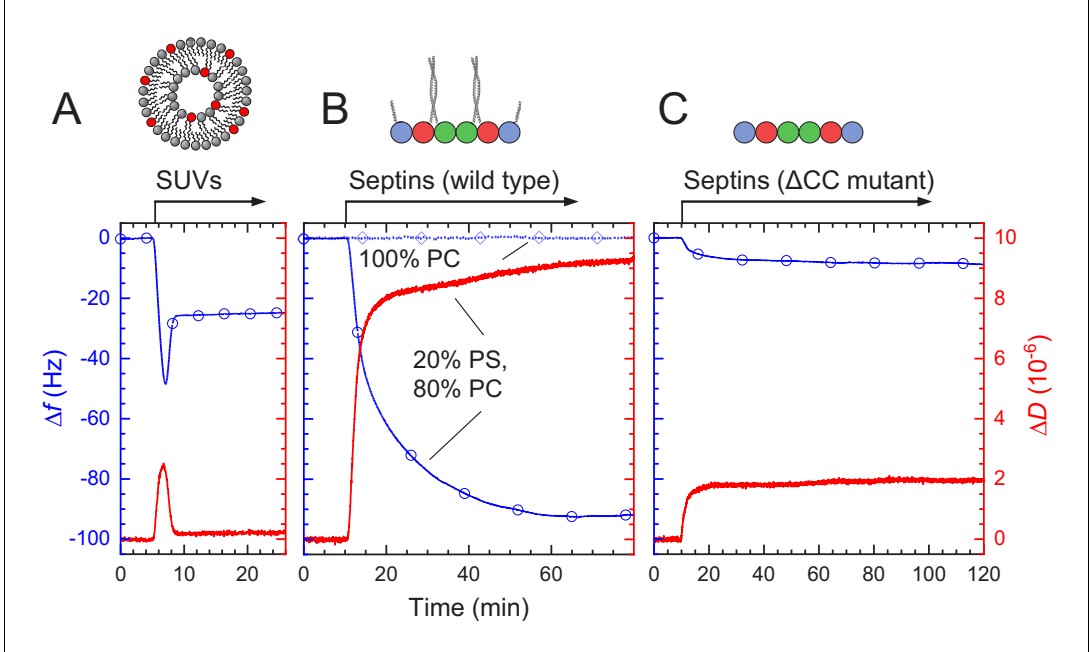

**Figure 7.** QCM-D measurements of fly septin hexamer adsorption on supported lipid bilayers (SLBs) containing anionic lipids show that septins form thin, rigid films. Shown are frequency shifts ($\Delta f$ – blue lines with symbols) and dissipation shifts ($\Delta D$ - red solid lines). Arrows on top of the graphs indicate start and duration of sample perfusion; during remaining times, plain buffer (A – vesicle buffer; B and C – septin polymerization buffer) was flown over the sensor surface. (A) Small unilamellar vesicles (SUVs at 50 µg/mL; 20% DOPS, 80% DOPC) were exposed to a plain silica surface to form a SLB. (B–C) Hexamers of wild type septin (B) and the ΔCC septin truncation mutant (C) (60 nM each), were exposed to 80% PC/20% PS SLBs (solid line with circle symbols). (B) also shows control frequency data for exposure of wild type septin (100 nM) to pure DOPC SLBs (dotted line with diamond symbols).

The online version of this article includes the following figure supplement(s) for figure 7:

**Figure supplement 1.** Lipid composition dependence of septin adsorption.
**Figure supplement 2.** Coverage-dependent ΔD/-Δf ratios for fly septins.

(*Reviakine et al., 2011*), and for globular proteins the major source of compliance typically are flexible hinges linking the proteins to the surface or inter-connecting protein domains (*Reviakine et al., 2011*; *Johannsmann et al., 2009*). Membrane-bound septins attained dissipation over frequency shift ratios $\Delta D$/-$\Delta f$ between $(0.20 \pm 0.01) \times 10^{-6}$/Hz at low coverage ($\Delta f = -15 \pm 5$ Hz) and $(0.09 \pm 0.01) \times 10^{-6}$/Hz at high coverage ($\Delta f < -90$ Hz; $N = 4$; *Figure 7—figure supplement 2*). For comparison, monolayers of streptavidin attain $\Delta D$/-$\Delta f$ ratios of $<0.01 \times 10^{-6}$/Hz when physisorbed on gold, and between $0.08 \times 10^{-6}$/Hz (at low coverage) and $0.015 \times 10^{-6}$/Hz (at high coverage) when linked via biotins on a short, flexible linker to a supported lipid bilayer (*Johannsmann et al., 2009*). The higher $\Delta D$/-$\Delta f$ ratios for septins are consistent with a high degree of flexibility in the linkage between the coiled coils and the globular domains of septin hexamers. Indeed, $\Delta D$/-$\Delta f$ ratios close to $0.1 \times 10^{-6}$/Hz have been reported for monolayers of neutravidin with short (i.e. several ten base pairs long) double-stranded DNA strands grafted to it (*Tsortos et al., 2008*), and for monolayers of streptavidin with linear oligosaccharide chains grafted to it (*Thakar et al., 2014*). Possibly the compliance of the coiled coils themselves, and/or the linkage between septins and the membrane, also contribute to the elevated $\Delta D$/-$\Delta f$ ratio relative to fully globular proteins.

## Septin's coiled coil domains are essential for forming multilayers

The data described above collectively demonstrate that fly septin hexamers form filaments that interact with each other both in the plane of the membrane, and out-of-plane. Which septin domains mediate these interactions? One candidate are the C-terminal coiled coils. In prior work on yeast septin octamers, coiled coils have already been proposed to be involved in septin filament pairing

and, on membranes, in the formation of perpendicular octamer cross-bridges between two paired filaments (*Bertin et al., 2008*; *Bertin et al., 2010*). Consistent with this view, a recent structural study of human septin coiled-coils employing X-ray crystallography, NMR, and modeling indicated that the coiled coils can form both parallel dimers that stabilize septin filaments and antiparallel dimers that form cross-bridges (*Leonardo et al., 2021*). A second candidate are the G-domains, which have been proposed to be involved in lateral interactions among yeast septin filaments on membranes (*Bertin et al., 2010*).

In order to test the arrangement of septin hexamers in the presence of G-domain interactions alone, we generated coiled-coil truncated fly septin hexamers (ΔCC mutant). Transmission EM confirmed that fly septins with their C-termini truncated from all subunits form stable hexamers (*Figure 1—figure supplement 1B*), similarly to C-terminally truncated human septin hexamers (*Sirajuddin et al., 2007*). EM imaging showed that in low salt polymerization buffer, the ΔCC mutant only formed short rods (*Figure 8A–B*) with lengths varying continuously between ~20 and ~80 nm (*Figure 8—figure supplement 1*). However, on lipid monolayers the C-terminally truncated fly septin hexamers formed dense arrays of aligned filaments (*Figure 8C–F*). Thus, membrane-binding promotes filament formation, mirroring earlier observations for yeast septin octamers with C-terminally truncated Cdc11 subunits (*Bertin et al., 2010*). The width of the ΔCC septin filaments was on average slightly smaller than that of full length septin filaments, around 3.5 nm (*Figure 3—figure supplement 1B*), likely due to the absence of coiled coils. In places we can see three filaments side by side instead of two (indicated by triple blue lines in *Figure 8D*). The mutant filaments were even more closely spaced than wild type septin filaments, with a center-to-center distance for filaments within pairs of around 4.5 nm (*Figure 8G* and *Figure 3—figure supplement 1A*), consistent with direct contact between the globular domains deprived of the CC domain of adjacent filaments due to crowding and perhaps specific G-domain interactions.

The septin density on the lipid monolayers observed in EM images for the ΔCC mutant was notably higher than for the full-length septins, even though the solution concentrations were the same. This observation suggests that the ΔCC mutant perhaps has a reduced tendency to form multilayers. To test this hypothesis, we performed QCM-D measurements for the ΔCC mutant on bilayers containing 20% PS. As shown in *Figure 7C*, the ΔCC mutant binds to the bilayers but the frequency shift is much smaller than for the full-length septins, indicating a thinner layer. In this case, the -Δ$f$ values remained less than 10 Hz, consistent with a single septin monolayer. Thus, the EM and QCM-D data demonstrate that the coiled coils are not needed for membrane binding, but they are needed for filament pairing, the formation of cross-bridges, and the stacking of septins on top of each other.

## Discussion

### Species-dependence of septin-membrane interactions

We investigated the influence of membrane-binding on septin hexamer assembly by reconstituting recombinant animal septins on supported lipid bilayers and imaging septin assembly with several complementary techniques (see *Table 1* for summary of techniques and experimental conditions). Fluorescence imaging revealed that fly septin hexamers have a high affinity for negatively charged lipid bilayers, which competes with the septin–septin lateral interactions that prevail in bulk solution, and that they form dense membrane-associated meshworks. Electron microscopy (EM) imaging revealed that these meshworks are predominantly comprised of paired filaments and meshworks thereof. A similar organization was observed for mammalian septin hexamers, consistent with earlier findings for septin-containing porcine brain extracts (*Tanaka-Takiguchi et al., 2009*; *Yamada et al., 2016*). Finally, atomic force microscopy (AFM) of septins on lipid bilayers showed that septin filaments laterally associate into bundles but also stack on top of each other. AFM imaging, fluorescence recovery after photobleaching and QCM-D experiments indicate that the septins are immobile and firmly attached to the membrane. By contrast, C-terminally truncated septin hexamers form monolayers and the filaments do not form pairs and meshworks, showing that the coiled coils of septins are crucial for mediating septin-septin interactions.

Prior in vitro studies of septin-membrane binding focused mainly on mitotic, Cdc11-capped budding yeast septin octamers (*Bertin et al., 2010*; *Bridges et al., 2014*). We find several striking similarities between the membrane interactions of animal (fly and mammalian) septin hexamers and

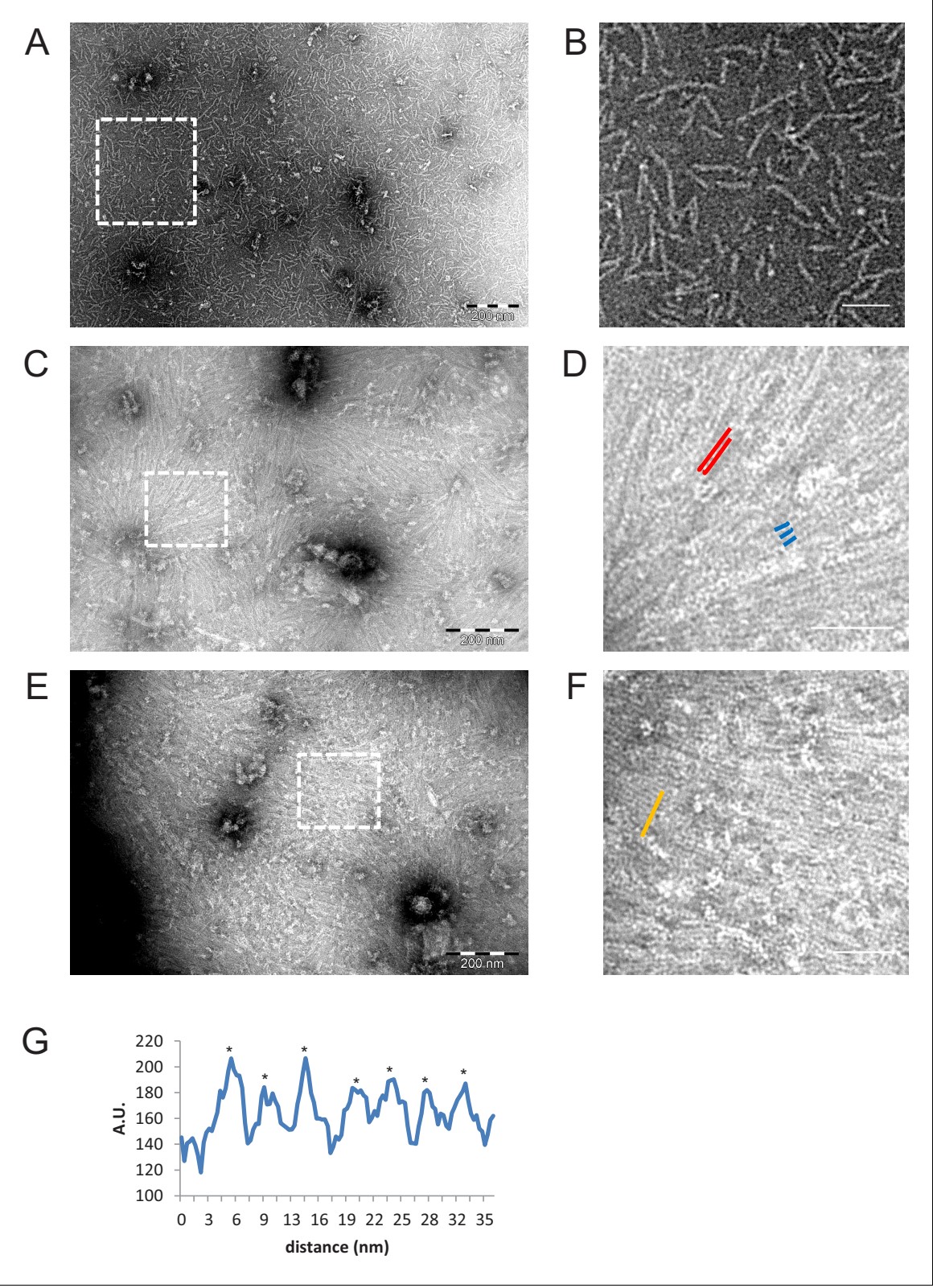

**Figure 8.** Negative stain EM images of C-terminally truncated fly ΔCC-septins. (**A**) ΔCC-septin hexamers in septin polymerization buffer. (**B**) Zoom-in of the boxed region in (**A**) shows that ΔCC-septin hexamers (84 nM) predominantly form monomers and short oligomers of hexamers with a length of ~24–60 nm. (**C**) ΔCC-septin hexamers (210 nM) on a negatively charged lipid monolayer composed of 80% DOPC, 10% PS, and 10% PI(4,5)P$_2$. (**D**) Zoom-in of the boxed region in (**C**) shows sheets of closely spaced, long filaments (red lines) and occasionally three filaments lined up side by side (trimer

*Figure 8 continued on next page*

*Figure 8 continued*

highlighted in blue). (E) Another example of the ΔCC-septin hexamers (210 nM) on a lipid monolayer composed of 80% PC, 10% PS, and 10% PIP(4,5)P$_2$. (F) Zoom-in of the boxed region in (E) again shows sheets of closely spaced, long, unpaired filaments. (G) Line scan (from area highlighted in panel F by an orange line) reveals a center-to-center (asterisks) spacing of ~4.5 nm. Scale bars are 200 nm (A,C,E) and 50 nm (B,D,F).

The online version of this article includes the following figure supplement(s) for figure 8:

**Figure supplement 1.** Lengths of ΔCC-septin hexamer filaments formed in low-salt septin polymerization buffer.

Cdc11-capped yeast septin octamers. First, membrane binding in both cases promotes septin polymerization. For fly septin hexamers, the threshold for polymerization is lowered more than 10-fold, from 200 nM in solution to less than 12 nM on negatively charged membranes (*Figure 9A*). For yeast septins, a similar enhancing effect was observed on membranes containing PI or PI(4,5)P$_2$ (*Bertin et al., 2010*; *Bridges et al., 2014*). The polymerization-enhancing effect of membrane binding is likely due to the increase of the effective septin concentration caused by 2D confinement. In addition, membrane binding may orient septins in a manner that promotes their polymerization, as seen for other proteins such as annexin A5 (*Oling et al., 2001*). A second similarity between the behavior of animal and yeast septins on membranes is that they both form paired filaments visible by transmission electron microscopy, with a narrow spacing of ~2 nm, much narrower than the ~10 nm spacing observed for yeast septin filaments formed in solution (*Bertin et al., 2008*). A third similarity is that paired filaments of both animal and yeast septins are interconnected by bridges visible by electron microscopy, which appear to be formed by short and thin (single) septin filaments having a length and axial spacing that correspond to the length of a single protomer (*Bertin et al., 2010*). Similar arrays have been observed by electron microscopy in yeast cells (*Ong et al., 2014*), but not yet in animal cells. Note that yeast septins were studied mostly by transmission electron microscopy only, which provides a 2D-projection. It is unknown whether yeast septins form multi-layers on membranes, as we see for fly septins by AFM and QCM-D.

Our work also reveals two striking differences between animal and yeast septins. The first difference is their polymerization in solution. Fly septins form thick and rigid bundles with tapered ends in solution, suggesting that the septins have strong lateral interactions that promote bundling. On membranes, however, we observed by TIRF microscopy a gradual transition from dilute arrays of thick bundles to dense arrays of paired filaments with increasing net surface charge. This observation suggests a competition of septin-membrane interactions with lateral septin-septin interactions, which suppresses the formation of thick bundles on membranes. By contrast, yeast septin octamers tend to form paired filaments in solution (*Bertin et al., 2008*; *Bertin et al., 2010*) (as confirmed by quantitative mass-mapping STEM in our study, see *Figure 1E*), although they do form bundles under certain conditions (*Booth and Thorner, 2016*). Thus, in solution fly septins have a stronger propensity for bundling than yeast septins, but on membranes the two septins behave similarly. The second striking difference between yeast and animal septins revealed here is their membrane binding selectivity. We find that fly septins form similar structures on membranes containing PS or PI(4,5)P$_2$, and that the main determinant of fly septin binding and filament organization in both cases is the net surface charge of the membrane. By contrast, Cdc11-capped yeast septin octamers were shown to be highly selective for PI, PI(4)P, PI(5)P and PI(4,5)P$_2$, while they did not interact with PS (*Khan et al., 2018*; *Bridges et al., 2016*; *Bertin et al., 2010*; *Beber, 2018*; *Cannon et al., 2019*; *Bridges et al., 2014*; *Casamayor and Snyder, 2003*; *Onishi et al., 2010*). The origin of the differences in lipid specificity remains unknown. It is thought that septins interact with negatively charged phospholipids *via* a polybasic region close to the N-terminus that is composed of a sequence of 1–7 basic amino acids (*Zhang et al., 1999*; *Casamayor and Snyder, 2003*). This stretch is very similar to polybasic sequences found in gelsolin, profilin, G-protein-coupled receptor kinases and ion channels (*Zhang et al., 1999*), which are all reported to interact with PI(4,5)P$_2$. In the future, perhaps molecular dynamics simulations can identify the determinants of lipid selectivity (*Lee et al., 2014*).

It is noteworthy that the intricate, highly organized septin filament meshworks observed at the membrane of dividing budding yeast (*Bertin et al., 2012*; *Ong et al., 2014*) have not been documented to date in animal cells. Although we cannot exclude that such assemblies exist in animal cells, this difference could be due to inherent differences between yeast and animal septins at the

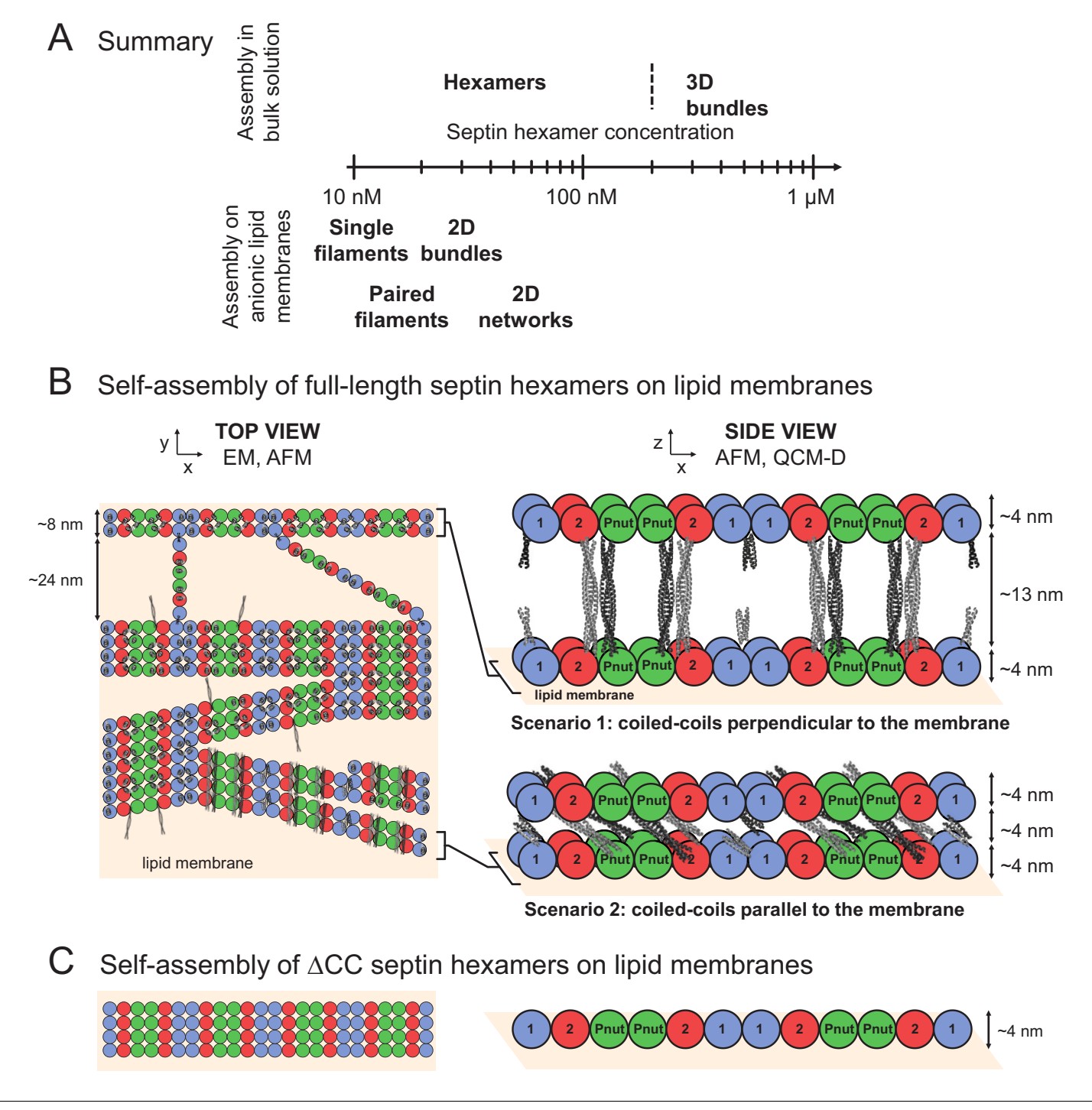

**Figure 9.** Model of membrane-templated septin self-assembly. (**A**) Diagram summarizing the regimes of fly septin self-assembly observed in the solution phase (top) and on anionic lipid membranes (bottom). (**B–C**) Left column: top views of the membrane proximal septin layer. Right column: side views of all septin layers. The coiled-coils are depicted to scale using available crystal-structures of coiled-coils (see Materials and methods) in light-gray for coiled coils emanating from the bottom septin layer and dark-gray for coiled-coils from the top layer. (**B**) Full-length septin hexamers form meshworks of paired filaments on the membrane, which are bridged by hexamers and dimers thereof and form bundles with filaments joining or emerging at a shallow angle. A second layer of septin filaments is recruited by coiled-coil interactions oriented vertically (scenario 1) or in-plane (scenario 2). For simplicity, scenario two is depicted with coiled coils oriented perpendicular to the filament axis though they may in reality adopt a range of orientations. (**C**) ΔCC-septin hexamers lacking coiled coils form a monolayer of filaments interacting by excluded volume interactions, perhaps augmented by G-domain interactions. Dimensions indicated in the schematic are based on the known size of septin subunits from EM and calculated

*Figure 9 continued on next page*

Figure 9 continued

lengths of the coiled-coil regions, and are consistent with the measured filament widths and spacings in-plane (from EM images) and with the measured height of the septin films (from AFM and QCM-D).

molecular level. Despite the homology between yeast and animal septins (on the order of 25–44% sequence identity), the end subunits of the respective septin protomers, SEPT2/DSep1 and Cdc11/ Shs1 in human/fly and yeast, respectively, differ considerably in their C-terminal extensions. Cdc11/ Shs1 have C-termini that are 2–3 times longer than their animal counterparts (118 and 211 residues for Cdc11 and Shs1 vs 55 and 57 residues for SEPT2 and DSep1, respectively). Cdc11 and Shs1 termini are predicted to form coiled-coils twice as long as the ones of SEPT2/DSep1, and with flexible hinges between the α6 helix and the coiled-coils 2–3 times longer than their animal counterparts (*Figure 1B,C* and *Taveneau et al., 2020*). Given the above differences between yeast and animal septins, the very high conservation between fly and mammalian septins (on the order of 65–75% identity), and the ubiquitous presence and essential functions of animal septins, studying animal septin assembly promises to advance our understanding of how differences between yeast and animal septins lead to differences in their organization and function.

## Towards a model of septin assembly on membranes

What do the imaging experiments collectively teach us about the assembly mechanism of fly septin hexamers on lipid membranes? A firm conclusion is difficult because there are still a number of important unknowns. It is still under debate where the membrane-binding region is located (*Cannon et al., 2019*; *Castro et al., 2020*; *Omrane et al., 2019*). We observe that fly septin hexamers are recruited to the membrane by electrostatic interactions with anionic lipids (PS and/or PI(4,5) $P_2$). It is generally believed that the N-terminal polybasic domains of septins interact with the membrane, which would leave the coiled coils free to rotate in the half-space above the membrane, although a recent study on yeast septin octamers suggested that membrane binding is aided by an amphipathic helix motif identified on the C-terminal side of the coiled-coil of Cdc12 (*Cannon et al., 2019*). In our case, however, we observe that C-terminally truncated septins still bind membranes and our data on full-length septins are consistent with the coiled coils being available for interactions with coiled coils on neighboring septins. Since the coiled coils are connected to the body of the septin complex *via* unstructured regions that can act as hinges, we assume that the coiled coils can rotate in the entire half-plane above the membrane, so they are available for both in-plane and out-of-plane interactions with coiled coils on neighboring septin subunits/filaments. The nature of these interactions is still unknown (i.e. assembly of two *vs.* four coiled coils, and parallel *vs.* antiparallel assembly). A recent structural study of human septin coiled coils indicated that these can form both parallel dimers that stabilize septin filaments and antiparallel dimers that form cross-bridges (*Leonardo et al., 2021*). Nevertheless, our imaging and QCM-D experiments do allow us to present a speculative model of membrane-templated septin self-assembly, which is illustrated in *Figure 9*.

On the left in *Figure 9B*, we display the proposed organization of septin filaments in the membrane-proximal layer. The hexamers form paired filaments by a combination of end-to-end association mediated by homodimerization of the terminal DSep1 subunits of adjacent hexamers, and lateral associations. Paired filaments in turn form in-plane bundles that occasionally branch and merge. In addition, paired septin filaments are bridged by hexamers or dimers thereof. Similar bridges were observed for yeast septin octamers on lipid monolayers, where the bridge length and spacing were also multiples of the octamer length (*Bertin et al., 2010*). In the context of yeast septins, it was proposed that coiled coils mediate septin filament pairing and the formation of perpendicular octamer cross-bridges between paired filaments (*Bertin et al., 2008*; *Bertin et al., 2010*). The interpretation in that case was that at least some C-terminal extensions on the terminal Cdc11 subunits of paired filaments were free to engage in interactions with C-terminal extensions on the bridging filaments. We here propose that fly septins may similarly form bridges *via* interactions of the C-terminal coiled coils of the terminal subunits (DSep1), facilitated by their orientational flexibility conferred by the hinge regions. Indeed for the C-terminally truncated septin hexamers, both pairing and cross-bridging are abrogated (*Figure 9C*).

**Table 1.** Overview of the analysis technique and assay conditions for each figure, summarizing the septin concentration, anionic lipid content of the membrane, and type of membrane used (lipid monolayers in case of negative stain EM imaging, supported lipid bilayers (SLB) in case of TIRF, AFM, and QCM-D, and large unilamellar vesicles (LUV) in case of cryoEM imaging).

| Figure | Assay | Technique | Septin concentration [nM] | Anionic lipid content |
|---|---|---|---|---|
| **Figure 1** | WT-fly septin in solution | TIRF and STEM | 100, 200, 500 | *No membrane present* |
| Suppl 1 | Fly septin in solution (WT and ΔCC mutant[*]) | TEM | 65 nM (wild type) 84 nM (ΔCC mutant) | *No membrane present* |
| Suppl 2 | WT-fly septin in solution | STEM | 500 | *No membrane present* |
| **Figure 2** | WT-fly septin on SLB | TIRF | 1000 | 1) 1–8 mole% PI(4,5)P$_2$ 2) 5, 10, 20% mole% PS |
| Suppl 1 | WT-fly septin on SLB | TIRF | 10, 100, 150, 200, 300, 500 | 5, 10, 20% mole% PS |
| Suppl 2 | WT-fly septin on SLB | FRAP | 500 | 20% PS |
| **Figure 3** | WT-fly septin on monolayer | TEM | 65 | 1) 20% PS 2) 5% PI(4,5)P$_2$, 3) 20% PS and 5% PI(4,5)P$_2$ |
| Suppl 1 | WT-fly septin on monolayer | TEM | 65 | Same as **Figure 3** |
| Suppl 2 | Mammalian septin on monolayer | TEM | 70 | 20% PS and 5% PI(4,5)P$_2$ |
| **Figure 4** | WT-fly septin on LUVs | cryoEM | 160 | 5% PI(4,5)P$_2$ |
| Suppl 1 | WR-fly septin on LUVs | cryoEM | 160 | 5% PI(4,5)P$_2$ |
| Suppl 2 | WT-fly septin on LUVs | cryoEM | 160 | 5% PI(4,5)P$_2$ |
| *Figure 4—video 1* | WT-fly septin on LUVs | cryoEM | 160 | 5% PI(4,5)P$_2$ |
| **Figure 5** | WT-fly septin on SLB | AFM | 12, 24, 60 | 20% PS |
| Suppl 1 | WT-fly septin on SLB | AFM | 12, 24, 60 | 20% PS |
| Suppl 2 | WT-fly septin on SLB | AFM | 60 | 20% PS |
| Suppl 3 | WT-fly septin on SLB | AFM | 24,60 | 20% PS |
| Suppl 4 | WT-fly septin on SLB | AFM | 60 | 20% PS |
| **Figure 6** | WT-fly septin on SLB | AFM | 60 | 20% PS |
| **Figure 7** | Fly septin on SLB (WT and ΔCC) | QCM-D | 60 | 20% PS |
| Suppl 1 | WT-fly septin on SLB | QCM-D | 60 | 1) 20% PS 2) 5% PI(4,5)P$_2$, 3) 20% PS and 5% PI(4,5)P$_2$ |
| Suppl 2 | WT-fly septin on SLB | QCM-D | 60 | 20% PS |
| **Figure 8** | ΔCC fly septin in solution and on monolayer[*] | TEM | 84, 210 | 10% PS + 10% PIP(4,5)P$_2$ |
| Suppl 1 | ΔCC fly septin in solution | TEM | 84 | *No membrane present* |

[*]We matched the weight concentrations for ΔCC-septin hexamers and full-length septin hexamers in EM experiments in high-salt solutions (0.02 mg/mL). Due to the different molecular weights of these hexamers, this corresponds to slightly different molar concentrations (i.e. 84 nM for ΔCC-septins and 65 nM for full-length septins).

To explain why septins form multi-layers, we hypothesize that not all coiled coils are engaged in interactions within the first membrane-proximal septin layer, thus leaving some coiled coils free to engage off plane. Pairing of the septin filaments in the membrane plane is very tight, with a spacing of only 2 nm, much smaller than the 13 nm extended length of the long coiled coils on Pnut and DSep2 subunits. Similarly tight pairing was previously observed for yeast septins on lipid monolayers (*Bertin et al., 2010*). To explain this, we consider that the coiled-coils may be oriented either upwards and engage transversely with coiled coils on a second layer of septins (*Figure 9B* right, scenario 1), or parallel to the membrane sandwiched between two layers of septins (*Figure 9B* right, scenario 2). According to scenario 1, we would expect a layer height of 21 nm, which is larger than the maximum height of ~18 nm seen by AFM for dense yet sub-saturated membranes, and the average heights in the range of 13–19 nm seen by QCM-D (depending on membrane composition) for

close-to-saturated membranes. According to scenario 2, we would expect a layer height of ~12 nm, in better agreement with the typical heights seen in AFM. However, the AFM images show that the bundle heights are variable. We therefore anticipate a combination of both scenarios, or even alternate scenarios, given the length and orientational freedom of the coiled coils and the known ability of coiled coils to form both parallel and antiparallel multimers (*Leonardo et al., 2021*; *Woolfson, 2017*; *Lupas et al., 2017*). Importantly, layering is restricted to just two layers. Our interpretation is that coiled coils are mostly engaged in in-plane associations while fewer are engaged in the recruitment of a new layer of septins, so that with increasing layer height, there is a smaller and smaller probability of recruiting additional material on top. This ultimately generates septin arrays that are very extensive in-plane, but limited in thickness to a ~ 12-nm-thick array with a layer of coiled coils sandwiched between two layers of globular domains. We note that this model is valid under conditions where septin assembly is initiated at the membrane (i.e. at septin concentrations below 200 nM, consistent with physiological levels of cytoplasmic septins in the range of 100–200 nM measured in fungi [*Bridges and Gladfelter, 2016*]). Thicker layers appear to be rare, and may arise either from a different organization of the second septin layer or from an additional septin layer. To elucidate the exact roles of the coiled coils and distinguish between contributions from the short versus the long coiled coils, it could be interesting in the future to study mixtures of wild-type and ΔCC septin hexamers or to study additional septin mutants with selective removal of either the short or long coiled coils. Since AFM lacks chemical specificity, it would probably be necessary to use superresolution fluorescence microscopy to resolve the potential exclusion of ΔCC septins within a mixed septin assembly and to localize the coiled coils. In addition, computational modeling will be essential to model the contributions of the short and long coiled coils and the G-domains of septins to septin-septin and septin-membrane interactions.

## Biological implications

Our observations of septin filament networks on model biomembranes suggest the possibility that septins in cells may directly enhance the rigidity of the cell cortex by forming filaments on the inner leaflet of the plasma membrane. Our AFM data suggest that membrane-bound septin filaments are intrinsically stiff and ductile while the link to the membrane is fluid. It will be interesting to probe the mechanical stability of the membrane-septin composite envelope in the future by reconstituting septins on free-standing membranes such as giant unilamellar vesicles (*Beber, 2018*; *Beber et al., 2019*). With their ability to form rigid filaments and extended meshworks thereof, septins could furthermore have an important role in mechanical integration of the cell cortex, since the actin-myosin cortex in mesenchymal and epithelial cells is known to turn over rapidly (*Bovellan et al., 2014*; *Fritzsche et al., 2016*; *Fischer-Friedrich et al., 2016*; *Clément et al., 2017*). Although septins in cells are more dynamic than in vitro, with typical in vivo half-times for fluorescence recovery on the order of 100 s (depending on septin subunit), the turnover rate of septins is still slower than for actin (where the half-time for fluorescence recovery is on the order of tens of seconds) (*Hagiwara et al., 2011*). In the context of amoeboid T-cell migration, it has for instance been suggested that stable septin structures are important to ensure directional extension of leading-edge protrusions (*Tooley et al., 2009*). Reconstitution of animal septins in the combined presence of membranes and the actin/microtubule cytoskeleton promises to help explore the contribution of septins to cell cortex mechanics. New 3D-superresolution techniques (*Kanchanawong et al., 2010*) and EM tomography (*Bertin et al., 2012*) of the cell cortex may be able to resolve the proximity of cortical septins to the cell membrane to answer the question whether septins indeed interact directly with the membrane at the cell cortex. Cortical septins are associated with both flat and curved membrane regions in the cell. Here, we focused on flat membranes. It will be interesting to study whether animal septin binding and assembly depends on the degree of membrane curvature as in the case of yeast septin octamers, which preferentially bind membranes with a curvature in the micrometre range (*Bridges et al., 2016*; *Beber et al., 2019*; *Cannon et al., 2019*).

## Materials and methods

**Key resources table**

*Continued*

| Reagent type (species) or resource | Designation | Source or reference | Identifiers | Additional information |
|---|---|---|---|---|
| Reagent type (species) or resource | Designation | Source or reference | Identifiers | Additional information |
| Strain, strain background (*Escherichia coli*) | BL21(DE3) | Agilent Technologies | Agilent Technologies: 200131 | |
| Recombinant DNA reagent | pnEA-vH encoding His$_6$-DSep1 (plasmid) | PMID:24633326 | | |
| Recombinant DNA reagent | pnCS encoding DSep2 and Pnut-StrepII (plasmid) | PMID:24633326 | | |
| Recombinant DNA reagent | pnCS encoding mEGFP-DSep2 and Pnut-StrepII (plasmid) | PMID:27473911 | | |
| Recombinant DNA reagent | pnEA-vH encoding His$_6$-DSep1ΔC56 (plasmid) | this paper | | See Materials and methods section, *Septin purification* |
| Recombinant DNA reagent | pnCS encoding DSep2ΔC111 and PnutΔC123-StrepII (plasmid) | this paper | | See Materials and methods section, *Septin purification* |
| Recombinant DNA reagent | pnEA-vH encoding mouse SEPT2 (plasmid) | PMID:24633326 | | |
| Recombinant DNA reagent | pnCS encoding human SEPT6 and human SEPT7-StrepII (plasmid) | PMID:24633326 | | |
| Peptide, recombinant protein | *Drosophila melanogaster* septin hexamers composed of His$_6$-DSep1-, DSep2, Pnut-StrepII | PMID:24633326 | | |
| Peptide, recombinant protein | Coiled-coil truncation mutant of *Drosophila melanogaster* septin hexamers composed of His$_6$-DSep1ΔC56, DSep2ΔC111and PnutΔC123-StrepII | this paper | | See Materials and methods, Section *Septin purification* |
| Peptide, recombinant protein | *Drosophila melanogaster* septin hexamers composed of His$_6$-DSep1, mEGFP-DSep2, and Pnut-StrepII | PMID:27473911 | | |
| Peptide, recombinant protein | Mammalian septin hexamers composed of mouse His$_6$-SEPT2, human SEPT6, and human SEPT7-StrepII | PMID:24633326 | | |
| Peptide, recombinant protein | Budding yeast septin octamers composed of Cdc3, Cdc10, Cdc11, and His$_6$-Cdc12 | PMID:18550837 | | |
| Commercial assay, kit | HisTrap FF crude column | GE Healthcare | 17-5286-01 | |
| Commercial assay, kit | StrepTrap HP | GE Healthcare | 28-9075-46 | |
| Commercial assay, kit | 4–15% gradient Mini-PROTEAN TGX Precast Protein Gels | Bio-Rad | #4561084 | |
| Commercial assay, kit | Precision Plus Protein Kaleidoscope protein standard | Bio-Rad | 1610375 | |
| Biological Sample (*genus Tobamovirus*) | Tobacco mosaic virus (TMV) rods | *Godon et al., 2017* PMID:28017791 | | Gift from Dr. J.L. Pellequer (Institut de Biologie Structurale, Grenoble, France) |

## Septin purification

Chemicals were bought from Sigma unless indicated otherwise. Recombinant *Drosophila* septin hexamers composed of DSep1 with an N-terminal His$_6$ tag, DSep2, and Pnut with a C-terminal Strep tag II (WSHPQFEK, 1058 Da) were purified from *Escherichia coli* BL21(DE3) cells (Agilent Technologies) using a two-tag affinity purification scheme to aid the isolation of full-length complexes as

explained in *Mavrakis et al., 2016*. Using the same cloning strategy as for wild type complexes (*Jones et al., 1992*), a coiled-coil truncation mutant of fly septin hexamers referred to as the ΔCC mutant was made with C-terminal truncations right after the end of the α6-helix (DSep1ΔC56, DSep2ΔC111, PnutΔC123). Cell cultures for wild type and ΔCC septins were grown at 37°C and expression was induced with 0.5 mM isopropyl-β-D-1-thiogalactopyranoside (IPTG) when the optical density at 600 nm (OD600) reached a value between 2 and 3. Fluorescently tagged septin complexes were obtained by tagging DSep2 with monomeric enhanced green fluorescent protein (mEGFP) on its N-terminus (*Jones et al., 1992*). To ensure GFP-folding, cell cultures for GFP-labeled septins were grown at 37°C until the OD600 reached 0.6–0.8 and expression was induced overnight at 17°C.

Cell pellets collected after centrifugation (10 min, 2800 × g, 4°C) were lysed on ice with a tip soni-cator in *lysis buffer* (50 mM Tris-HCl pH 8, 500 mM KCl, 10 mM imidazole, 5 mM MgCl$_2$, 20 mM MgSO$_4$, 0.1 mM guanosine diphosphate (GDP), 5% glycerol) supplemented with 0.25 mg/mL lyso-zyme (5934-D, Euromedex), 1 mM phenylmethylsulfonyl fluoride (Sigma), Complete protease inhibi-tor cocktail (Roche), and 10 mg/L DNase I (Sigma). The clarified lysate obtained after centrifugation (30 min at 20000 × g, 4°C) was first run on a nickel affinity column (HisTrap FF crude column, GE Healthcare) equilibrated with *septin one buffer* (50 mM Tris-HCl pH 8, 500 mM KCl, 10 mM imidaz-ole, 5 mM MgCl$_2$, 5% glycerol) by eluting with 250 mM imidazole. The peak fractions were pooled and run on a streptavidin affinity column (StrepTrap HP, GE Healthcare) equilibrated with *septin two buffer* (50 mM Tris-HCl pH 8, 300 mM KCl, 5 mM MgCl$_2$, 5% glycerol) by eluting with 2.5 mM d-Des-thiobiotin. Pooled fractions from the elution peak were dialyzed against *storage buffer* (50 mM Tris-HCl pH 8, 300 mM KCl, 5 mM MgCl$_2$, 5% glycerol, 5 mM dithiothreitol (DTT)) at 4°C overnight and concentrated to ∼3–5 mg/mL using Vivaspin six concentrators (Sartorius, 6 ml, 100 kDa cut-off). Ini-tial purifications used an inverted order of columns and a gel filtration step (Superdex 200 HiLoad 16/60, GE Healthcare) instead of a dialysis step (*Mavrakis et al., 2014*; *Mavrakis, 2016*). Mamma-lian septin hexamers composed of mouse SEPT2 (98.6% identical to human SEPT2, differing in 5 out of 361 residues, based on sequence alignment with the Clustal Omega from https://www.ebi.ac.uk/Tools/msa/clustalo/) with an N-terminal His$_6$ tag, human SEPT6, and human SEPT7 with a C-terminal Strep tag II were purified in the same manner (*Mavrakis et al., 2014*). The SEPT7 isoform used was as annotated in *Macara et al., 2002*. Budding yeast septin octamers (composed of Cdc3, Cdc10, Cdc11, and His$_6$-Cdc12) were purified as described (*Bertin et al., 2008*) and stored in 50 mM Tris-HCl buffer (pH 8) supplemented with 300 mM KCl. Purified septin complexes were flash-frozen in liq-uid nitrogen and stored at −80°C.

## Septin characterization

The concentration of septin complexes was determined by measuring the optical absorbance of the solutions at 280 nm with a UV-VIS spectrophotometer (Thermo Scientific, Nanodrop 2000) and sub-tracting the contribution of scattering (typically ∼2–3%) from absorbance measurements between 320 and 340 nm (*Birdsall et al., 1983*). We used extinction coefficients of 0.545 L × g$^{-1}$ × cm$^{-1}$ for full-length *Drosophila* septins, 0.584 L × g$^{-1}$ × cm$^{-1}$ for mEGFP-tagged septins, 0.645 L × g$^{-1}$ × cm$^{-1}$ for the ΔCC mutant, and 0.565 L × g$^{-1}$ × cm$^{-1}$ for the mammalian septins, with the values all calculated from the sequence (*Gasteiger et al., 2003*). Molar concentrations were converted into weight concentrations using the calculated molecular weights of 306.9 kDa for full-length untagged *Drosophila* septins, 361.6 kDa for mEGFP-tagged septins, 237.8 kDa for the ΔCC mutant, and 285.7 kDa for the mammalian septins. The purity and integrity of the proteins was evaluated for each batch by sodium dodecyl sulfate polyacrylamide gel electrophoresis (SDS-PAGE) using 4–15% gradient gels (Mini-PROTEAN TGX Precast Protein Gels, Bio-Rad) with a Precision Plus Protein Kaleidoscope (Bio-Rad) protein standard (see *Figure 1—figure supplement 1A–B* for full-length and ΔCC septins, respectively). The length of the purified complexes (as a readout of the stability of the hexameric arrangement) was evaluated by transmission electron microscopy (TEM). Solutions of septins diluted to 65 nM with a buffer containing 300 mM KCl, to prevent polymerization, were deposited on glow-discharged carbon coated copper grids (CF300-Cu, Electron Microscopy Sciences) and incubated for at least 30 min. The samples were negatively stained with 2% uranyl acetate (Nanoprobes, Yaphank, USA), air-dried, and examined with a FEI Tecnai Spirit (120 kV) electron microscope (Ther-moFisher, Waltham, USA). We observed rod-shaped oligomers, mostly having the expected 24 nm length of a hexamer (*Figure 1—figure supplement 1A–B*). We performed a total of 10 preparations

of the full-length fly septins and verified that the protein quality by SDS-PAGE and by TEM analysis was constant. We used 2 preparations of C-terminally truncated fly septins, 1 preparation of yeast septins, and 1 preparation of mammalian septins.

## Sequence alignment of fly and human septins

Sequence alignment of fly and human septins was performed using the Clustal Omega multiple sequence alignment program (https://www.ebi.ac.uk/Tools/msa/clustalo/) using the full-length sequences of these septins. The % identity scores mentioned in the text are extracted from the percent identity matrix output of the multiple sequence alignment. A score of 100% is given to two identical amino acids and a score of zero otherwise.

## Septin coiled-coil size prediction

To identify the regions of the C-termini of fly and human septins that can adopt a coiled-coil conformation, we employed the coiled-coil prediction algorithm COILS (https://embnet.vital-it.ch/software/COILS_form.html) using the full-length sequences of these septins. The output was identical when we used the scoring matrix MTIDK (derived from myosins, paramyosins, tropomyosins, intermediate filaments, desmosomal proteins and kinesins) or MTK (derived from the sequences of myosins, tropomyosins and keratins *Lupas et al., 1991*), and for both weighted and unweighted scans, indicating little bias due to the high incidence of charged residues. *Figure 1A* displays the predicted structures for fly septins starting right after the α6-helix (ending in ..DRLAK for DSep1; ..RLEQ for DSep2; ..KLSE for Pnut). For DSep1, the algorithm predicted a 28-residue-long coiled-coil (LGEKDR...AQMQAR) with 99.8% probability. An 86-residue-long C-terminal coiled-coil (QQTFEA.. QQLATA) was predicted for DSep2 (47% probability for the first 21 residues and 86–99.9% probability for the following 65 ones). An 86-residue-long stretch in the C-terminus of Pnut (LTQMEE... HVTLEE) was further predicted to adopt a coiled-coil conformation with >97.9% probability. *Figure 1B* shows the corresponding predictions for human septins starting right after the α6-helix (ending in .. RLKR for SEPT2; ..KLEE for SEPT6; ..KLAA for SEPT7). The C-terminus of human SEPT2 was predicted to form a 28-residue-long coiled-coil (LLEKEA...QMQMQG) with 90.2% probability. The coiled-coil prediction for mouse SEPT2 is identical. An 86-residue-long stretch in the C-terminus of SEPT6 (QETYEA...TAAELL) was predicted to adopt a coiled-coil conformation with 74–100% probability. An 86-residue-long C-terminal coiled-coil (LAQMEE..RILEQQ) was further predicted for SEPT7 with 99.5–100% probability. The coiled-coil predictions were fully in line with the α-helix prediction output of the secondary structure prediction programs PHYRE2 and PSIPREDv4.0 (*Kelley et al., 2015*; *Buchan and Jones, 2019*). From the end of the a6-helix to the start of the predicted coiled-coils, there are stretches of 24, 15, and 15 residues for DSep1, DSep2, and Pnut, respectively, and stretches of 15, 15, and 19 residues for SEPT2, SEPT6, and SEPT7, respectively, that are predicted to be unstructured. To depict septin coiled-coils, the crystal structures of which have not yet been published, in *Figure 1C*, *Figure 1—figure supplement 1A*, *Figure 7B* and *Figure 9*, we have used the available crystal structures of coiled-coil dimers and tetramers from vimentin (PDB 3UF1) (coiled-coil side-views in the figures) and the early endosomal SNARE complex (PBD 2NPS) (coiled-coil top-views in the figures). The molecular structure visualization program PyMOL was used for isolating helical segments, and rotating and coloring the dimeric and tetrameric coiled-coil segments shown in the images.

## Preparation of small unilamellar lipid vesicles

Lipids were purchased from Avanti Polar Lipids (Birmingham, AL). The following lipids were bought and stored at a concentration between 10 and 25 mg/mL in chloroform: 1,2-dioleoyl-*sn*-glycero-3-phosphocholine (PC), 1,2-dioleoyl-*sn*-glycero-3-phospho-L-serine (PS), and 1,2-dioleoyl-*sn*-glycero-3-phosphoethanolamine-N-(lissamine rhodamine B sulfonyl) (rhodamine-PE). Furthermore, 1,2-dioleoyl-*sn*-glycero-3-phospho-(1'-myo-inositol-4',5'-bisphosphate) (ammonium salt) (PI(4,5)P$_2$) was bought in powder form and stored as a 0.5 mg/mL solution in a chloroform:methanol:water mixture (20:9:1 vol ratio). Small unilamellar vesicles (SUVs) were prepared by drying mixtures of PC/PS/rhodamine-DOPE (79.7:20.0:0.3 molar ratio), PC/PI(4,5)P$_2$/rhodamine-PE (94.7:5.0:0.3), or PC/PS/PI(4,5)P$_2$/rhodamine-DOPE (74.7:20.0:5.0:0.3) in glass vials using an air stream. PC/PS lipid mixtures were resuspended in a filtered and degassed imidazole buffer of pH 7.4 (20 mM imidazole-HCl, 50 mM

KCl, and 2 mM MgCl$_2$). The total lipid concentration of the stock solutions was 0.5 mM for TIRF imaging and 2.5 mM for atomic force microscopy (AFM) and quartz crystal microbalance with dissipation monitoring (QCM-D) experiments. PI(4,5)P$_2$-containing lipid mixtures were resuspended at a total lipid concentration of 0.25–0.5 mM in a 50 mM citrate buffer of pH 4.8 containing equimolar amounts of trisodium citrate and citric acid, 50 mM KCl, and 0.1 mM ethylenediaminetetraacetic acid (EDTA). The acidic pH promotes the formation of homogeneous and fluid bilayers by reducing the net charge on the head group of PI(4,5)P$_2$ from −4 to −3 (*Braunger et al., 2013*). SUVs for TIRF experiments were obtained by sonication with a tapered microtip (Branson, USA; 3 mm diameter) for 30 min in pulsed mode (30 s on/30 s off at 10% amplitude). SUVs for AFM and QCM-D experiments were obtained by exposing the lipid suspensions to five freeze/thaw cycles followed by sonication with a microtip (FisherBrand; 2 mm diameter) for 30 min in pulsed mode (1 s on/1 s off at 30% amplitude) with ice cooling. The SUVs were centrifuged (15 min at 15000 × *g*) to remove any titanium particles that might come off the sonicator tip, along with residual lipid aggregates, although these were minimal as no white pellet could be observed after sonication. All vesicles were stored in the fridge. PI(4,5)P$_2$-containing SUVs were used within 5 days, and all other vesicles within 30 days. We confirmed by QCM-D measurements that SLB formation and septin binding were unchanged over the course of these storage periods.

## Fluorescence microscopy

Samples were prepared in home-made flow channels assembled from cleaned coverslips and microscope slides that were rendered hydrophilic in base piranha (5% hydrogen peroxide, 5% ammonium hydroxide in water) at 70˚C. Flow channels were prepared by assembling a dried coverslip and microscope slide with two parafilm spacers spaced apart by ~2 mm and fixed in place by melting on a hotplate at 120˚C. We used either 100 mol-% mEGFP-tagged hexamers or mixtures of 90 mol-% untagged hexamers with 10 mol-% mEGFP-tagged hexamers (see captions). Septin assembly on supported lipid bilayers (SLB) was performed on the microscope. SLBs were first formed by flushing the SUV stock solution into the flow channels and allowing for SUV rupture and spreading (10 to 15 min). Correct SLB formation was ascertained by checking that the fluorescence signal was spatially uniform and, using fluorescence recovery after photobleaching measurements, that the bilayer was fluid. Residual SUVs were removed by washing the SLBs with five channel volumes of septin *polymerization buffer* (20 mM imidazole pH 7.4, 1 mM DTT, 0.1 mM MgATP, 50 mM KCl and 2 mM MgCl$_2$), containing 2 mM trolox to suppress blinking (*Rasnik et al., 2006*) and a 2 mM protocatechuic acid/ 0.1 µM protocatechuate 3,4-dioxygenase mixture to suppress photobleaching (*Aitken et al., 2008*). Septin hexamers diluted to a concentration between 10 nM and 1 µM in *polymerization buffer* were then flushed onto the SLBs and assembly was allowed to proceed for 30 min at 20˚C unless mentioned otherwise. In case of net-neutral (pure PC) bilayers, we had to include 0.1 wt-% methylcellulose in the buffer to crowd the septins close enough to the surface to allow visualization by TIRF microscopy. This methylcellulose concentration is low enough such that it does not cause filament bundling (*Köhler et al., 2008*). In case of anionic bilayers, we did not use methylcellulose to ensure that any surface recruitment was really due to membrane binding.

Most TIRF images were obtained with a Nikon Eclipse Ti-E inverted microscope equipped with a TI-TIRF-E motorized TIRF Illuminator (Roper Scientific), a Nikon Apo TIRF 100×/1.49 N.A. oil immersion objective, a 491 nm/50 mW Calypso laser (Cobolt, Solna, Sweden), and a QuantEM 512SC EMCCD camera (Photometrics, Roper Scientific, Tucson, AZ, USA) using an exposure time of 50 ms. This microscope was controlled with MetaMorph 7.5 software (Molecular Devices, Sunnyvale, CA, USA). TIRF images in *Figure 2—figure supplement 1C* were obtained using a Nikon Ti2-E microscope equipped with a Gataca iLAS2 azimuthal TIRF illumination system, 488 nm laser (Gataca laser combiner iLAS2), and Andor iXon Ultra 897 EM-CCD camera using an exposure time of 75 ms. Images were processed (contrast enhancement and look-up table inversion) and analyzed (integrated intensity calculations) with Fiji software (*Schindelin et al., 2012*). Fluorescence recovery after photobleaching was performed using a NikonA1 confocal microscope using a 100× oil immersion objective and an argon laser (Coherent, CA, USA) at excitation wavelengths of 488 nm for septins and 561 nm for rhodamine-PE lipids. The septins or the bilayer were first bleached by briefly (1 s) illuminating a circular region with a radius of 5 µm using a high (70%) output laser power. The fluorescence recovery was then measured for 5–7 min by illuminating the sample using a low (0.1–2%) output laser power and 30 ms exposure time. For the first 30 s after the bleach, images were

acquired at one frame per second. For the remaining acquisition time, images were acquired at a rate of 1 frame per 10 s.

## Scanning transmission electron microscopy of septin bundles

To determine the morphology and mass per length (MPL) of septin bundles formed in solution, we performed scanning transmission electron microscopy (STEM) using tobacco mosaic virus (TMV) rods with a well-defined length (300 nm), width (18 nm), and MPL (131 kDa/nm) as an internal mass standard (*Freeman and Leonard, 1981*; *Sousa and Leapman, 2013*). Carbon or formvar+carbon-coated copper grids (Ted Pella, Redding, CA, USA) were first incubated for 30 s with 3 µL TMV (0.01–0.02 mg/ml dispersion in phosphate-buffered saline *Godon et al., 2017*), a kind gift from Dr. Jean-Luc Pellequer (Institut de Biologie Structurale, Grenoble, France), washed with ultrapure water, and blotted with filter paper. Next, 5 µL of a solution of septins preassembled for 1 hr at 20°C at a concentration between 50 and 500 nM was deposited and left for 1 min. Finally, the grids were washed with ultrapure water to remove excess salts and left to dry in air at 37°C. Images of 3072 × 2207 pixels (16 bits) were acquired at different magnifications (from 15,000 to 50,000×) and at an acceleration voltage of 10–20 kV, a current of 100 pA, and a pixel dwell time of 3–5 µs on a FEI Verios 460 STEM microscope. The MPL was determined by analysis in Fiji *Schindelin et al., 2012* following a published procedure (*Sousa and Leapman, 2013*). Briefly, we selected images that contained both TMV rods and septin bundles. For intensity calibration, boxes with a width of 25–45 nm (depending on the image resolution) and a length of 100 nm were drawn along the TMV rods, to obtain the protein signal, and on both sides of each rod, to obtain the background signal. The conversion factor from intensity (in counts) to mass (in kDa) per pixel was determined from the integrated intensities of at least three TMV rods per image. A similar procedure was then used to obtain the MPL of septin bundles, drawing 100 nm long boxes over the bundles that encompassed their width (20–50 nm), together with equal-sized boxes on either side of the bundle for background subtraction. The width of the septin bundles was measured by drawing a line in Fiji perpendicularly across septin bundles and estimating the distance between the edges by eye (see examples in *Figure 1—figure supplement 2B,C*).

## Transmission electron microscopy of septins on lipid monolayers

Samples for transmission electron microscopy (TEM) were prepared by incubating septin hexamers with lipid monolayers formed at an air-buffer interface. Teflon wells were filled with 20 µL of a solution of 65 nM (fly) or 70 nM (mammalian) septin hexamers in a 50 mM Tris-HCl (pH 8) buffer with 50 mM KCl. A drop (~0.5 µL) of a 0.5 g/L lipid solution in chloroform was deposited at the air-water interface in each well to form a lipid monolayer. The Teflon block was kept overnight at 4°C in a humid chamber, while stirring the solution in each well with a magnetic stirrer bar. Lipid monolayers with adsorbed septins were collected by briefly placing hydrophobic grids (Electron Microscopy Sciences, Hatfield, PA, USA) on the surface of the solutions with the carbon-coated side facing the solution. Grids with the collected monolayers facing up were stained at once with 2% uranyl formate (Electron Microscopy Sciences) in water by depositing 4 µL on the grid and simultaneously blotting the excess solution. Samples were imaged using a Tecnai Spirit electron microscope (Thermo Fischer) operated at an acceleration voltage of 120 V. We repeated the experiments twice having each time two replicates. Filament width and distance measurements were performed manually using Image J, based on line profiles.

## Cryo-electron microscopy of septins on large unilamellar lipid vesicles

Samples for cryoEM were prepared by incubating septin hexamers with large unilamellar vesicles (LUVs). Lipid mixtures at a total lipid content of 50 µg per sample composed of 85 mol-% PC, 5 mol-% PI(4,5)P$_2$, and 10 mol-% PE (i.e. phosphatidylethanolamine) were dried in glass vials with argon and left in vacuum for 2 hr to remove chloroform. We included PE because earlier studies for yeast septin octamers showed that PE favors septin-membrane binding (*Beber, 2018*). We did not observe any qualitative changes in the appearance of the fly septin filaments on LUVs made with or without PE present (data not shown). LUVs were formed by hydrating the lipids in high salt buffer at a concentration of 0.5 mg/mL (20 mM imidazole pH 7.4, 1 mM DTT, 300 mM KCl and 2 mM MgCl$_2$). The vesicles were next incubated for 30 min with septins at a concentration of 160 nM (0.05 g/L) in

high-salt septin storage buffer, corresponding to a lipid-to-protein weight ratio of 10:1. Meanwhile, lacey carbon coated grids (Electron Microscopy Science, EMS, France) were plasma treated for 30 s to hydrophilize the surface and remove any impurities. Vesicles diluted in septin *polymerization buffer* to a final concentration of 0.3 µM (0.1 g/L) were applied (*Beber et al., 2019*) to the grid straight after plasma treatment. Prior to the vitrification of the sample, 10 nm gold beads (EMS) were added in solution to be subsequently used as fiducial markers. Excess liquid was blotted and the samples were vitrified in liquid ethane using an automatic plunge freezer (EM PG, Leica, Wetzlar, Germany). Samples were imaged using a FEI Tecnai G2 microscope equipped with a LaB6 filament and a 4K × 4K CMOS camera F416 (TVIPS, Gauting, Germany) operated at an acceleration voltage of 200 kV and a magnification of 50000×. Data collection was carried out using the EMTool (TVIPS) software suite. Tilt series were acquired from −60 to 60 degrees using a saxton angular data collection scheme. Individual images were collected with 0.8 electrons per Å$^2$ for a total dose of less than 70 electrons per Å$^2$. IMOD software (*Kremer et al., 1996*) was primarily used for data processing and alignment for individual images. The reconstructions were performed using either IMOD (Weighted back projection) or Tomo3D (SIRT) (*Agulleiro and Fernandez, 2011*). The segmentation of the volumes was performed manually using IMOD. We repeated the experiments three times. Filament width and distance measurements were performed manually using Image J, based on line profiles.

## Atomic force microscopy of septins on supported lipid bilayers

AFM samples were prepared on silicon substrates in order to approximate the conditions used for TIRF imaging and QCM-D measurements. Silicon wafers (Prime Silicon, Les Ulis, France) were cut into squares of 9 × 9 mm$^2$, rinsed with absolute ethanol, blow-dried with N$_2$ gas, and treated with UV/ozone (Bioforce Nanoscience, Ames, IA) for 30 min to remove organic contaminants and render the surface hydrophilic. Each substrate was glued (Picodent Twinsil, Wipperfürth, Germany) on a 15 mm diameter metal disc (Agar Scientific) covered with a hydrophobic protector film (Bytac surface protection laminate; Sigma). An SLB was formed on the silicon wafer piece by incubating for 30 min with 100 µL of a 100 µg/mL SUV solution diluted in septin *polymerization buffer* with an enhanced KCl concentration to promote SUV rupture (20 mM imidazole pH 7.4, pH 7.4, 1 mM DTT, 150 mM KCl and 2 mM MgCl$_2$). Residual SUVs were washed off with 1 mL of septin *polymerization buffer*. The SLB was kept hydrated by leaving 100 µL buffer on the wafer surface and 50 µL of septin solution was added to reach a final concentration of either 12, 24, or 60 nM. After 15 min incubation at room temperature (19°C), unbound protein was washed off with 1 mL of *polymerization buffer*. In most experiments, the samples were fixed for 1 min with one wt-% glutaraldehyde (GTA) in *polymerization buffer* and washed. Unfixed septins had a similar morphology as fixed septins but were difficult to image at high resolution because they were easily disrupted by the scanning AFM tip. Reproducibility across biological replicates was checked by performing experiments on five protein preps.

AFM images were acquired at room temperature (23°C) with a Nanoscope Multimode 8 system (Bruker, CA, USA) on samples immersed in *polymerization buffer*. Images of various lateral dimensions (5–20 µm) at a resolution of 512 × 512 pixels and a scan rate of 0.8 Hz were recorded with silicon cantilevers (ScanAsyst-Fluid+; Bruker) with a nominal spring constant of 0.7 N/m and a sharp pyramidal tip with a nominal radius of 2 nm and a maximal radius of 12 nm. Images were recorded in Peak Force Tapping mode with a typical driving frequency of 2 kHz and tapping amplitude of 50 nm. Images were second-order flattened using open source software Gwyddion (*Nečas and Klapetek, 2012*) to correct for sample tilt and curvature of the *xy* piezo scanner. Analysis of septin thread heights and widths was performed by constructing height profiles (perpendicular to the thread axis, and averaging five pixels along the axis) in Gwyddion software. Heights were defined as the peak value above the flat membrane surface. Widths were measured at a height of 1 nm below peak height. Sections across narrow threads generally showed a single peak, but wider threads (>30 nm in width) often showed multiple peaks. In the latter case, widths were measured at 1 nm below the height of the two outermost peaks. We recall that tip convolution effects lead to an enlarged effective width. Considering the expected tip shape, and the height at which the widths were measured, we expect tip convolution to entail a width overestimation by between 2 and 10 nm.

## QCM-D measurements of septin-membrane binding

To analyze the kinetics of septin binding and self-organisation on lipid membranes, we employed a surface analytical technique known as quartz crystal microbalance with dissipation monitoring (QCM-D). This is an acoustic technique that measures the adsorption of biomolecules onto the surface of a piezoelectric quartz crystal that is cyclically sheared at its acoustic resonance frequency by applying an alternating current (*Ward and Buttry, 1990*). The adsorption of molecules to the sensor surface results in a frequency shift $\Delta f$ that, to a first approximation, is proportional to the acoustic mass (including hydrodynamically coupled solvent) of the adsorbed film (*Sauerbrey, 1959*) and a dissipation shift $\Delta D$ that provides information on the mechanical properties of the film (*Johannsmann et al., 2009*; *Eisele et al., 2012*).

We used silica-coated QCM-D sensors (QSX303, Biolin Scientific, Sweden) that were cleaned in an aqueous solution of 2% sodium dodecyl sulfate (30 min), rinsed with ultrapure water, blow-dried with $N_2$ gas, and exposed to UV/ozone (BioForce Nanosciences, Ames, IA) for 30 min. The sensors were then immediately mounted in the Q-Sense Flow Modules of the Q-Sense E4 system (Biolin Scientific) and experiments were started within 15 min. Experiments were performed at a working temperature of 23˚C. The instrument was operated in flow mode, meaning that sample solution was continuously perfused through the measurement chamber at a rate of 20 μL/min with a syringe pump (KD Scientific). We sequentially incubated the sensors with SUVs to form a bilayer and then septins, interspersed by rinsing steps. The surface was first equilibrated with *vesicle buffer* before flowing in the SUV suspension. The adsorption of the SUVs onto the sensor surface and the subsequent formation of a supported lipid bilayer (SLB) were monitored by measuring $\Delta f$ and $\Delta D$. The frequencies are measured in odd multiples (overtones) of the fundamental resonance frequency of the piezoelectric sensor crystal. In our experiments, the fundamental resonance frequency was 5 MHz and data were recorded at overtones $i$ = 3 to 13. For further analysis, we selected data obtained at the 7th overtone, in view of its stable response across all measurements. Other overtones provided qualitatively similar information and are thus not shown. The frequency shifts were normalized according to $\Delta f = \Delta f_7/7$. Once $\Delta f$ and $\Delta D$ reached stable values, indicating that the SLB was ready, the surface was rinsed with *vesicle buffer* to remove residual SUVs and then with septin *polymerization buffer* to equilibrate the ionic conditions. Next, the sensor was perfused with 60 nM septins in *polymerization buffer* for at least 60 min. Finally, we tested the reversibility of septin-membrane binding by rinsing the sensor with polymerization buffer. The experiments were performed at least in duplicate. Reproducibility across biological replicates was checked by performing experiments on five protein preps.

QCM-D measurements can provide information on the thickness of the adsorbed septin films. Provided that the septin layer is sufficiently uniform, dense, and rigid, the Sauerbrey equation (*Sauerbrey, 1959*) relates the frequency shift to the areal mass density of the film, $m_{\text{QCM-D}}$, which equals the layer density $\rho_{\text{QCM-D}}$ times the layer thickness $d_{\text{QCM-D}}$:

$$\Delta f = -C^{-1} \times m_{\text{QCM-D}} = -C^{-1} \times \rho_{\text{QCM-D}} \times d_{\text{QCM-D}} \tag{1}$$

The mass sensitivity constant $C$ is 18 ng/cm²/Hz for QCM-D sensors with a fundamental resonance frequency of 5 MHz and the mass density for protein films is 1.1 g/cm³ to a good approximation (*Reviakine et al., 2011*). For sufficiently dense and rigid ($\Delta D$/-$\Delta f$ << 0.4 × $10^{-6}$/Hz) films, the film thickness $d_{\text{QCM-D}}$ is therefore proportional to $\Delta f$ with a proportionality constant of 6.1 Hz/nm.

## Acknowledgements

We thank Marjolein Kuit-Vinkenoog and Jeffrey den Haan for protein purification, Cristina Martinez Torres for discussions on the AFM data analysis and help with STEM imaging, Aditya Iyer for help with STEM imaging, and Anders Aufderhorst-Roberts for help with AFM imaging. We also thank the Cell and Tissue Imaging (PICT-IBiSA), Institut Curie, member of the French National Research Infrastructure France-BioImaging (ANR10-INBS-04). AS and GHK gratefully acknowledge support by AMOLF, whose research program is part of the Netherlands Organisation for Scientific Research (NWO). GHK acknowledges financial support from the ERC (Starting Grant 335672; MINICELL) and the 'BaSyC - Building a Synthetic Cell' Gravitation grant (024.003.019) of the Netherlands Ministry of Education, Culture and Science (OCW) and the Netherlands Organization for Scientific Research

(NWO). RPR and FB gratefully acknowledge support from the AFM facilities of the Molecular and Nanoscale Physics Group (University of Leeds) and funding from the ERC (Starting Grant 306435; JELLY) and the BBSRC (Equipment grant BB/R000174/1). MM and AB gratefully acknowledge support by the Institut Curie, Institut Fresnel, and the Centre National de la Recherche Scientifique (CNRS) and funding from the Agence Nationale pour la Recherche (ANR grants ANR-13-JSV8-0002-01; SEPTIME and ANR-17-CE13-0014; SEPTIMORF) and the Fondation ARC pour la recherche sur le cancer (grant PJA 20151203182).

## Additional information

### Funding

| Funder | Grant reference number | Author |
|---|---|---|
| H2020 European Research Council | ERC StG 335672 MINICELL | Gijsje H Koenderink |
| H2020 European Research Council | ERC StG 306435 JELLY | Fouzia Bano<br>Ralf P Richter |
| Biotechnology and Biological Sciences Research Council | Equipment grant BB/R000174/1 | Ralf P Richter<br>Fouzia Bano |
| Agence Nationale de la Recherche | ANR-13-JSV8-0002-01 | Aurélie Bertin |
| Agence Nationale de la Recherche | ANR-17-CE13-0014 | Manos Mavrakis |
| Fondation ARC pour la Recherche sur le Cancer | PJA 20151203182 | Manos Mavrakis |
| Nederlandse Organisatie voor Wetenschappelijk Onderzoek | 024.003.019 | Gijsje H Koenderink |

The funders had no role in study design, data collection and interpretation, or the decision to submit the work for publication.

### Author contributions

Agata Szuba, Conceptualization, Data curation, Formal analysis, Validation, Investigation, Visualization, Methodology, Writing - original draft, Writing - review and editing; Fouzia Bano, Data curation, Formal analysis, Validation, Investigation, Visualization, Methodology, Writing - review and editing; Gerard Castro-Linares, Conceptualization, Data curation, Formal analysis, Validation, Investigation, Writing - review and editing; Francois Iv, Investigation, Methodology; Manos Mavrakis, Conceptualization, Resources, Supervision, Funding acquisition, Investigation, Visualization, Methodology, Writing - review and editing; Ralf P Richter, Aurélie Bertin, Conceptualization, Resources, Data curation, Software, Formal analysis, Supervision, Funding acquisition, Validation, Investigation, Visualization, Methodology, Writing - review and editing; Gijsje H Koenderink, Conceptualization, Resources, Data curation, Supervision, Funding acquisition, Investigation, Methodology, Writing - original draft, Project administration, Writing - review and editing

### Author ORCIDs

Fouzia Bano (iD) https://orcid.org/0000-0003-0634-7091
Ralf P Richter (iD) https://orcid.org/0000-0003-3071-2837
Aurélie Bertin (iD) https://orcid.org/0000-0002-3400-6887
Gijsje H Koenderink (iD) https://orcid.org/0000-0002-7823-8807

### Decision letter and Author response

Decision letter https://doi.org/10.7554/eLife.63349.sa1
Author response https://doi.org/10.7554/eLife.63349.sa2

## Additional files

### Supplementary files
• Transparent reporting form

### Data availability
All data generated or analysed during this study are included in the manuscript and supporting files.

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
