## [Decision Letter]

**Acceptance summary:**

This is an interesting study on reconstitution of *Drosophila* of septin hexamer filaments on supported bilayers and characterization of polymer biochemical and biophysical properties. The work extends some findings from budding yeast septins, but delves deeper into the molecular properties of metazoan septins, which appear to function differently from the yeast counterparts (e.g. in exactly how they relate to the cytokinetic actomyosin ring during ring constriction). The use of nanoscale imaging is also an important advance from this work.

**Decision letter after peer review:**

Thank you for submitting your article "Membrane binding controls ordered self-assembly of animal septins" for consideration by *eLife*. Your article has been reviewed by 3 peer reviewers, and the evaluation has been overseen by Mohan Balasubramanian as the Reviewing Editor and Olga Boudker as the Senior Editor. The following individual involved in review of your submission has agreed to reveal their identity: Virgile Viasnoff (Reviewer #1).

The reviewers have discussed the reviews with one another and the Reviewing Editor has drafted this decision to help you prepare a revised submission.

Summary:

The referees agree your work on reconstitution of *Drosophila* septin hexamers into filaments on supported bilayers and their characterization and comparison with the yeast counterparts is interesting and important. However, the referees raise a number of important points, all of which need to be addressed satisfactorily before publication.

Essential revisions:

1. In all sets of experiments different amounts of PS, PIP2 and septin concentrations are used. How can the obtained data be discussed in terms of these parameters, if they are not really comparable? What is the rationale of using the different conditions? For example: mEGFP-tagged fly septins are crowed with methylcellulose on neutral PC SLBs. Septin concentrations of 100-500 nM were used (Figure1). In Figure 2, however, 1000 nM septin is used without methylcellulose. AFM and QCM-D data were obtained with 20 % PS, no PIP2, and 10 nM septin concentrations. These conditions do not resemble the conditions in the TIRF experiments (as written). In the TIRF experiments 1000 nM septin was used. Cryo-EM data were obtained for 6 mol% PIP2, no PS.

In the discussion a model (or several models) are proposed, which appear to be highly speculative. The results are compared to those with yeast septins reported in literature. As this comparison is the major point that is made in the manuscript it would be important to perform at least one of the experiments with yeast septin for direct comparison.

2. The authors should create lipid bilayers on curved surfaces like glass rods to simulate the ability of the septins to create annulus structures as in vivo. Indeed it seems that on vesicles the structure of septin filaments hardly differ from the monolayer case. Adding topographica and geometrical cues to the septin assembly can potentially bring new insights in how they can assemble in rings or sheets.

3. Examinations of septin hexamers only containing the short or long coiled coils or mixing two populations of septin hexamers (wt and ΔCC) to see whether the ΔCC are excluded from any filament stacks would be highly recommended to support the final model.

4. Regarding the results in Figure 2D: A simple calculation explains why you find dense septin packing on SLBs at 10nM. Assuming a septin hexamer has the area of 4x24 nm2 and the flow chamber has an area of 5x20 mm2 and a height of 2mm, you would need about 1ꞏ 1012 septin hexamers to cover the SLB. This number of septins in the volume of the flow chamber would correspond to a concentration of about 8.7 nM. It is not clear why the authors did not check lower septin hexamer concentrations as this would simply require further dilution of the stock solution. These results seem to be also in conflict with the AFM results, where individual septin filaments are observed at 12nM and 24nM. The authors should clarify this difference.

5. The comments about mechanical stability of the lipid bound septins are unsurprising and not very conducive as they describe GTA fixed septins. Studies of lateral stability don't have to relate directly to enhanced cortex stability. It would have been more powerful to compare the stability of septin decorated GUVs.

Sorry, but the discussion about septin layer height limitation is very speculative and would be much better founded if the authors would have done some more experiments. The claim that the layer is self-limiting is not in line with the TIRF data that shows a steady increase of fluorescence intensity at 500nM septin. It would have been good to add AFM and QCM data on samples with higher septin concentrations, e.g. 500nM, to prove that the layer remains indeed within 12-21nm. It would also be insightful to either test mixtures of ΔCC septins with full length septins or to generate septins only lacking the long coiled-coils or the short coiled-coils to support the conclusions of the authors.

---

## [Author Response]

Essential revisions:1. In all sets of experiments different amounts of PS, PIP2 and septin concentrations are used. How can the obtained data be discussed in terms of these parameters, if they are not really comparable? What is the rationale of using the different conditions? For example: mEGFP-tagged fly septins are crowed with methylcellulose on neutral PC SLBs. Septin concentrations of 100-500 nM were used (Figure1). In Figure 2, however, 1000 nM septin is used without methylcellulose. AFM and QCM-D data were obtained with 20 % PS, no PIP2, and 10 nM septin concentrations. These conditions do not resemble the conditions in the TIRF experiments (as written). In the TIRF experiments 1000 nM septin was used. Cryo-EM data were obtained for 6 mol% PIP2, no PS.

We used a substantially wider septin concentration range in the TIRF experiments compared to the cryoEM/AFM/QCM-D experiments because TIRF was aimed at rapidly screening conditions, whereas cryoEM/AFM/QCM-D was aimed at subsequent detailed structural characterization for selected conditions. We see in retrospect that the logic behind this parameter choice may have been obscured by the complex narrative.

To aid the reader, we now summarize the parameters in a Table in the Supplementary Information and we better explain the logic behind the parameter choices in the Results section of the manuscript. In summary, the rationale for the parameter choices was as follows:

1. Septin concentrations: From the TIRF experiments screening a septin concentration range of 10-1000 nM, we concluded that septins start to bundle in solution above a threshold concentration of 200 nM (Figure 1D). At concentrations above 200 nM and in the presence of an adhesive membrane containing PS or PI(4,5)P_2_, we observed that bundles in solution co-exist with membrane-bound septins (see Author response image 1). To ensure that septin polymerization in EM/AFM/QCM-D experiments occurred at the membrane, we therefore selected a narrower concentration range of 10-70 nM for these experiments. The only exception is the cryoEM assay, where septins were incubated at a concentration of 160 nM with LUVs, because their positive curvature disfavors septin binding [3].

2. Lipid compositions: TIRF was used to perform a first rapid screen for the dependence of septin binding on anionic lipid content and to test for lipid binding specificity (with lipid concentration ranges of 1-8 mole% PI(4,5)P_2_; 5-20 mole% PS; 20% PS + 5% PI(4,5)P_2_; 10% PS + 10% PI(4,5)P_2_). Negative stain EM was used to test whether the anionic lipid species (PS versus PI(4,5)P_2_) affects the organization of membrane-bound septin filaments. Based on the combined results from TIRF and EM, we concluded that binding and self-assembly of septin hexamers depends on the net negative charge of the membrane but not on the anionic lipid species (TIRF in Figure 2B,C and EM in Figure 3). We chose 20% PS for all subsequent AFM and QCM-D experiments employing SLBs because earlier literature had established that the approximately even inter-leaflet distribution of PS lipids in liposomes is preserved upon SLB formation on silica substrates [4], whereas the effect of the solid support on the inter-leaflet distribution of PI(4,5)P_2_ has not yet been reported and, given the high charge on the head group of PI(4,5)P_2_ lipids, there is a risk that the inter-leaflet distribution may be uneven. Prompted by essential revision #5, we performed new QCM-D experiments for additional lipid compositions that matched the TIRF and EM conditions (5% PI(4,5)P2, and 20% PS + 5% PI(4,5)P2). These QCM-D measurements confirmed that membrane binding does not qualitatively change with anionic lipid species (Figure 5—figure supplement 3C and Figure 5—figure supplement 4B).

3. Methylcellulose: this crowding agent was used only in TIRF experiments, and even then only in cases where septins were assembled in the absence of a membrane (Figure 1) or in the presence of a non-adhesive membrane (0% PS condition in Figure 2C). Methylcellulose was needed in these cases to push the septin bundles towards the coverslip so they were visible by TIRF microscopy. In all the experiments where the bilayer contained anionic lipids, we did not add methylcellulose, in order to ensure that the septins observed by TIRF were truly membrane-bound.

Actions:

– We now explain the choice of parameters in the revised Results section each time we introduce a new experimental assay.

– We added a table to the Supplementary Information that summarizes the parameters for each figure. We corrected one mistake in the septin concentration used in cryoEM experiments (Figure 4 and its supplements), which was 160 nM (0.05 g/L) rather than 200 nM.

– We added a clarification to the caption of Figure 2, explaining that methylcellulose was present as a crowding agent for the 0% PS condition in Figure 2C but not in any of the other conditions.

– Prompted by essential revision #5, we performed new QCM-D experiments for additional lipid compositions that matched the TIRF and EM conditions (new Figure 7—figure supplement 1 in the revised manuscript).

**Author response image 1. respfig1:** TIRF imaging shows co-existence of bundles with membrane-bound septins when fly septin hexamers (100% GFP-tagged) are incubated at a concentration of 300 nM with a supported lipid bilayer containing 80% DOPC and 20% DOPS. (A) Image taken in TIRF mode showing membrane-bound septins (laser angle ~80°). (B) Off-TIRF-angle image (laser angle ~3°) showing bundle-like structures in the solution above the bilayer. Arrows point out examples of bundles. Images are contrast-inverted.

In the discussion a model (or several models) are proposed, which appear to be highly speculative. The results are compared to those with yeast septins reported in literature. As this comparison is the major point that is made in the manuscript it would be important to perform at least one of the experiments with yeast septin for direct comparison.

The major aim of our work was to study the influence of membrane binding on the self-assembly of animal (fly and mammalian) septins, which is interesting in its own right because it can help guide future work disentangling the many roles of septins in human (patho)physiology. It was not our explicit aim to perform a comparative study of animal and yeast septins. Nevertheless, since yeast septins are the most well-studied septins, we naturally compare our findings to available literature data for yeast septins in the Discussion section. Yeast septin self-assembly, both in solution [5, 6] and in the presence of lipid monolayers/bilayers [7-14], is already thoroughly documented in the literature, so we do not feel it is necessary to include any new data on yeast septins in our manuscript except for the STEM data in Figure 1E. The rationale for showing those STEM data is that our manuscript is the first instance where STEM is applied for quantitative mass mapping of septins so we felt it was useful to include yeast septins as a benchmark.

2. The authors should create lipid bilayers on curved surfaces like glass rods to simulate the ability of the septins to create annulus structures as in vivo. Indeed it seems that on vesicles the structure of septin filaments hardly differ from the monolayer case. Adding topographica and geometrical cues to the septin assembly can potentially bring new insights in how they can assemble in rings or sheets.

We agree that the specific question of the effect of membrane curvature on septin assembly is a very interesting one, which has so far been addressed only in the context of yeast septins in seminal studies of the Gladfelter [11, 13] and Bertin [3] labs. However, answering this question would necessitate a dedicated project to adequately address the interplay between septins and curvature. Such work is outside the scope of the current manuscript, whose aim was specifically to characterize septin-membrane binding in the *absence* of curvature to understand functions of septins in the cortex of animal cells.

Our manuscript nevertheless does include one set of experiments where septins were incubated with curved membranes, namely in the cryoEM experiments where septins were incubated with LUVs. However, the motivation for using LUVs was not to study curvature effects, but to examine septin binding on lipid *bilayers*, which are more physiologically relevant than the lipid monolayers required for negative stain TEM. As pointed out by the reviewer, we find similar paired filament arrangements on LUVs as on lipid monolayers. However, this finding does not exclude the possibility that membrane-binding and polymerization of fly septins might be curvature dependent, as was found for yeast septins [3, 11, 13].

Action: We more explicitly state in the revised Abstract that the scope of the paper is limited to septin interactions with non-curved membranes, we extended the Discussion section with a brief outlook towards future reconstitution studies of animal septins on curved substrates, and we refined the discussion of possible effects of curvature in the cryoEM data for LUV-bound septins.

3. Examinations of septin hexamers only containing the short or long coiled coils or mixing two populations of septin hexamers (wt and ΔCC) to see whether the ΔCC are excluded from any filament stacks would be highly recommended to support the final model.

We believe that the experimental support for the model presented in Figure 9 based on wild type (wt) and C-terminally truncated (∆CC) septin hexamers is sufficiently strong, since both the AFM and the QCM-D experiments clearly show from the comparison of wt and ∆CC that the coiled coils are necessary for out-of-plane stacking. Prompted by the reviewers’ comment, we did decide to strengthen experimental support for the model by demonstrating the generality of our findings by adding QCM-D data obtained for membranes of different lipid compositions (see Figure 7—figure supplement 2 in the revised manuscript; see second part of essential revision #5 for a further explanation).

Further detailed studies of the exact positioning of the short and long coiled coils are definitely interesting, but beyond the scope of our study. We anticipate that experiments mixing wild-type and ∆CC septin hexamers might be difficult to interpret given that AFM lacks chemical specificity so, combined with its relatively low spatial resolution, it is unlikely to resolve any potential exclusion of ΔCC septin hexamers within a mixed septin assembly. Super-resolution fluorescence imaging might be better suited since it offers chemical specificity [15]. In addition, computational modelling will be essential to dissect the roles of the short and long coiled coils. The high similarity in terms of C-terminal extension length and predicted coiled-coil formation among fly and mammalian septins (Figure 1A,B), and their difference from yeast septins, in particular from the end subunit Cdc11/Shs1, likely reflects the importance of the length of C-terminal extensions, flexible hinges and coiled-coils in septin filament organization. Thus, engineering of artificially short or long coiled coils for all septins in the hexamer might result in modes of organization that are not relevant to the organization of wild-type animal septins.

Action: In the Discussion section where we discuss the model in Figure 9, we now discuss possible follow-up studies to understand the molecular details of the roles of the short and long coiled coils in septin-septin and septin-membrane interactions.

4. Regarding the results in Figure 2D: A simple calculation explains why you find dense septin packing on SLBs at 10nM. Assuming a septin hexamer has the area of 4x24 nm2 and the flow chamber has an area of 5x20 mm2 and a height of 2mm, you would need about 1ꞏ 1012 septin hexamers to cover the SLB. This number of septins in the volume of the flow chamber would correspond to a concentration of about 8.7 nM.

The AFM images show that the septin coverage is actually quite sparse at 12 nM (Figure 5). To evaluate whether this is expected, we can calculate an upper limit for the amount of bound protein by assuming mass-transport limited binding from a still (and semi-infinite bulk) solution to a planar surface: Γ≤2cDtπ  [16]. With an incubation time *t* = 30 min, a septin concentration *c* = 10 nM, and diffusion constant *D* ≈ 20 μm^2^/s (a rough estimate for molecules with the size of septin hexamers in aqueous solution; consistent with fluorescence correlation spectroscopy measurements for yeast octamers [8] with *D* ≈ 31±5 μm^2^/s), this gives Γ≈ 0.20 pmol/cm^2^. With a footprint of 4 nm × 24 nm per septin hexamer, the maximal surface coverage thus is 12%. The septin hexamer surface coverage at 12 nM concentration is thus indeed expected to be rather low, consistent with our AFM data.

It is not clear why the authors did not check lower septin hexamer concentrations as this would simply require further dilution of the stock solution. These results seem to be also in conflict with the AFM results, where individual septin filaments are observed at 12nM and 24nM. The authors should clarify this difference.

The TIRF data at low septin hexamer concentration (10 nM) in Figure 2D of the original manuscript showed a clear fluorescence signal indicative of membrane-mediated septin recruitment, but unfortunately lacked the resolution required to resolve the precise arrangement or surface coverage of septins.

Prompted by the reviewer’s questions, we performed new experiments to test whether it is possible to visualize the septin filaments by going to low concentrations. We first performed experiments at septin hexamer concentrations of 10, 50 and 200 nM, using a mixture of 10% GFP-labeled hexamers and 90% unlabeled hexamers (Author response image 2). The fluorescence signal was rather weak and we were unable to distinguish septin filaments at the lowest concentration of 10 nM. At 50 nM and 200 nM we could, with some difficulty, distinguish septin filaments having a length of ~1.3 μm, consistent with typical lengths observed by AFM. To increase the signal, we raised the fraction of GFP-labeled hexamers to 100% and performed experiments on a range of septin hexamer concentrations between 3 and 300 nM (Author response image 3). At low concentrations (3-50 nM), we observed dark (i.e., bright) puncta indicative of protein aggregation, suggesting that the GFP tag interferes with normal septin hexamer assembly. Remarkably, when we increased the septin concentration to values of 100 nM and higher, we observed the appearance of a fibrillar meshwork of septin filaments that resembled the meshworks seen when only 10% of hexamers was labeled (Author response image 2 and Figure 2 in the original manuscript), but the filaments were more clearly visible due to the higher labeling degree. There appeared to be fewer aggregates; perhaps, at elevated septin concentration and in the presence of a membrane to template polymerization, the GFP tag is less perturbative. We regret to conclude that we are unable to obtain clear TIRF images of the fly septins at low septin concentrations, so we cannot make a direct quantitative comparison between the surface coverage in TIRF vs AFM images.

Action:

We now discuss the concentration-dependent surface coverage observed by AFM with estimates for mass-transport-limited binding in the Results section of the revised manuscript.

We replaced the earlier TIRF images showing the effect of septin concentration by newly acquired data (Figure 2—figure supplement 1C of the revised manuscript).

**Author response image 2. respfig2:** TIRF microscopy images of GFP-tagged fly septin hexamers (10% labelled hexamers, 90% dark hexamers) on a supported lipid bilayer containing 80% DOPC and 20% DOPS for a range of septin hexamer concentrations as labelled. Note that the image contrast is inverted. Scale bars are 5μm.

**Author response image 3. respfig3:** TIRF microscopy images of GFP-tagged fly septin hexamers (100% labelled) on a supported lipid bilayer containing 80% DOPC and 20% DOPS for a range of septin hexamer concentrations as labelled. Note that image contrast is inverted. Scale bars are 5μm.

5. The comments about mechanical stability of the lipid bound septins are unsurprising and not very conducive as they describe GTA fixed septins. Studies of lateral stability don't have to relate directly to enhanced cortex stability. It would have been more powerful to compare the stability of septin decorated GUVs.

Our data on the lateral stability of dense (60 nM) septin arrays in Figure 5—figure supplement 3C of the original manuscript indeed corresponded to conditions where septins were fixed with 1% GTA. We agree with the reviewers that one cannot draw conclusions about mechanical stability based on fixed samples. However, we did in fact obtain clear evidence of mechanical stability of arrays and single fibrils also for unfixed septin filaments (see Figure 5—figure supplement 3C and Figure 5—figure supplement 4B) and we now include these data in the revised manuscript. We also agree with the reviewers that lateral stability of the septin filaments does not have to relate directly to enhanced cortex stability and have therefore carefully rephrased our conclusions on implications for the mechanical stability of the cortex.

Action:

– We include the new AFM data for unfixed septin samples in the revised manuscript, showing them side-by-side with corresponding data for fixed septin samples. Revised Figure 5—figure supplement 3C demonstrates that dense arrays of fixed as well as unfixed septins are stable (data Author response image 1), while revised Figure 5—figure supplement 4 demonstrates that both fixed and unfixed septin filaments can be displaced intact across the SLB membrane (data from Author response image 1).

– We refined our discussion in the ‘Biological implications’ section of the Discussion on the possible role of septin filaments in controlling the mechanical stability of the cell cortex.

Sorry, but the discussion about septin layer height limitation is very speculative and would be much better founded if the authors would have done some more experiments. The claim that the layer is self-limiting is not in line with the TIRF data that shows a steady increase of fluorescence intensity at 500nM septin. It would have been good to add AFM and QCM data on samples with higher septin concentrations, e.g. 500nM, to prove that the layer remains indeed within 12-21nm.

In retrospect we realize that the TIRF data, which were merely intended for an initial broad sweep of parameter conditions, not for detailed quantification of binding, may have created some confusion. The concentration dependence of the TIRF signal does not translate directly in a density of bound protein because it includes both membrane-bound septins and solution phase septins present within the TIRF excitation field. We therefore quantified septin-membrane binding instead with QCM-D, which measures specifically the mass of surface-bound septins.

Our conclusion that septin layer growth is self-limiting is thus based on the QCM-D data, in particular on the time traces for the frequency shift in response to continuous flushing of the membranes with septins: Even though we continue to flush in septins into the channel at a constant concentration and at a constant flow rate, the membrane-bound layers always reached a limiting thickness.

In the original manuscript we only showed one example of this, on a 20% PS bilayer (Figure 7B). To strengthen our conclusions and show the generality for different lipid compositions, we now include new QCM-D data on membranes of different lipid compositions in the revised manuscript (Figure 7—figure supplement 2 in the revised manuscript). We note that our conclusions are valid for septin concentrations in the physiologically relevant regime, where septin assembly initiates at the membrane surface (concentrations <200 nM, see Figure 9A). The concentration of cytoplasmic septins in three different fungal model systems was measured by fluorescence correlation spectroscopy to be 100-200 nM [8], in line with the concentrations we used in our AFM and QCM-D experiments.

Action:

– We clarified both in the Results section where we present the QCM-D data and in the Discussion section where we discuss the model in Figure 9 that the time dependence of the frequency shift is proof of self-limiting layer growth.

– To reinforce this point, we now include new QCM-D data showing that film growth is self-limiting on membranes of different compositions (new Figure 7—figure supplement 1).

– We clarified that this conclusion holds under conditions where septin assembly is initiated at the membrane (<200 nM septin).

– To avoid confusion, we moved the TIRF data qualitatively showing the effect of septin concentration on membrane-binding from the main text to the supplementary data (Figure 2—figure supplement 1C), and we explain in the corresponding figure caption that the TIRF signal includes both membrane-bound and solution-phase septins.

It would also be insightful to either test mixtures of ΔCC septins with full length septins or to generate septins only lacking the long coiled-coils or the short coiled-coils to support the conclusions of the authors.

We refer to our response to essential revision #3 above.

References:

1. Macara, I.G., et al., Mammalian septins nomenclature. Mol Biol Cell, 2002. 13(12): p. 4111-3.

2. Hall, P., S. Russell, and J. Pringle, eds. The Septins. 2008, John Wiley: Chichester.

3. Beber, A., et al., Membrane reshaping by micrometric curvature sensitive septin filaments. Nat Commun, 2019. 10(1): p. 420.

4. Richter, R.P., N. Maury, and A.R. Brisson, On the effect of the solid support on the interleaflet distribution of lipids in supported lipid bilayers. Langmuir, 2005. 21(1): p. 299-304.

5. Bertin, A., et al., *Saccharomyces cerevisiae* septins: supramolecular organization of heterooligomers and the mechanism of filament assembly. Proc. Natl. Acad. Sci., 2008. 105(24): p. 8274-9.

6. Booth, E.A. and J. Thorner, A FRET-based method for monitoring septin polymerization and binding of septin-associated proteins. Methods Cell Biol., 2016. 136: p. 35-56.

7. Bertin, A., et al., Phosphatidylinositol-4,5-bisphosphate promotes budding yeast septin filament assembly and organization. J. Mol. Biol., 2010. 404(4): p. 711-31.

8. Bridges, A.A., et al., Septin assemblies form by diffusion-driven annealing on membranes. Proc Natl Acad Sci U S A, 2014. 111(6): p. 2146-51.

9. Beber, A., et al., Septin-based readout of PI(4,5)P2 incorporation into membranes of giant unilamellar vesicles. Cytoskeleton (Hoboken), 2018.

10. Casamayor, A. and M. Snyder, Molecular dissection of a yeast septin: distinct domains are required for septin interaction, localization, and function. Mol. Cell Biol., 2003. 23(8): p. 2762-77.

11. Bridges, A.A., et al., Micron-scale plasma membrane curvature is recognized by the septin cytoskeleton. J Cell Biol, 2016. 213(1): p. 23-32.

12. Khan, A., J. Newby, and A.S. Gladfelter, Control of septin filament flexibility and bundling by subunit composition and nucleotide interactions. Mol Biol Cell, 2018. 29(6): p. 702-712.

13. Cannon, K.S., et al., An amphipathic helix enables septins to sense micron-scale membrane curvature. J. Cell Biol., 2019. DOI: 10.1083/jcb.201807211.

14. Onishi, M., et al., Role of septins in the orientation of forespore membrane extension during sporulation in fission yeast. Mol Cell Biol, 2010. 30(8): p. 2057-74.

15. Kaplan, C., et al., Absolute Arrangement of Subunits in Cytoskeletal Septin Filaments in Cells Measured by Fluorescence Microscopy. Nano Lett, 2015. 15(6): p. 3859-64.

16. Hermens, W.T., et al., Effects of flow on solute exchange between fluids and supported biosurfaces. Biotechnol Appl Biochem, 2004. 39(Pt 3): p. 277-84.

17. Tanaka, T., M. Kinoshita, and K. Takiguchi, Septin-mediated uniform bracing of phospholipid membranes. Curr. Biol., 2009. 19(2): p. 140-5.

18. Adam, J., J. Pringle, and M. Peifer, Evidence for functional differentiation among Drosophila septins in cytokinesis and cellularization. Mol. Biol. Cell, 2000. 11(9): p. 3123-35.

19. Tada, T., et al., Role of Septin cytoskeleton in spine morphogenesis and dendrite development in neurons. Curr Biol, 2007. 17(20): p. 1752-8.

20. Xie, Y., et al., The GTP-binding protein Septin 7 is critical for dendrite branching and dendritic-spine morphology. Curr Biol, 2007. 17(20): p. 1746-51.

21. Kissel, H., et al., The Sept4 septin locus is required for sperm terminal differentiation in mice. Dev Cell, 2005. 8(3): p. 353-64.

22. Ihara, M., et al., Cortical organization by the septin cytoskeleton is essential for structural and mechanical integrity of mammalian spermatozoa. Dev Cell, 2005. 8(3): p. 343-52.

23. Kuo, Y.C., et al., SEPT12 orchestrates the formation of mammalian sperm annulus by organizing core octameric complexes with other SEPT proteins. J Cell Sci, 2015. 128(5): p. 923-34.

24. Taveneau, C., et al., Synergistic role of nucleotides and lipids for the self-assembly of Shs1 septin oligomers. Biochem J, 2020. 477(14): p. 2697-2714.

25. Sousa, A.A. and R.D. Leapman, Development and application of STEM for the biological sciences. Ultramicroscopy, 2012. 123: p. 38-49.

26. Sousa, A.A. and R.D. Leapman, Mass mapping of amyloid fibrils in the electron microscope using STEM imaging. Methods Mol Biol, 2013. 950: p. 195-207.

27. Mendonca, D.C., et al., A revised order of subunits in mammalian septin complexes. Cytoskeleton (Hoboken), 2019. 76(9-10): p. 457-466.

28. Soroor, F., et al., Revised subunit order of mammalian septin complexes explains their in-vitro polymerization properties. Mol Biol Cell, 2020: p. mbcE20060398.

29. DeRose, B.T., et al., Production and analysis of a mammalian septin hetero-octamer complex. Cytoskeleton (Hoboken), 2020. 77(11): p. 485-499.

30. Soroor, F., et al., Revised subunit order of mammalian septin complexes explains their in vitro polymerization properties. bioRxiv:doi.org/10.1101/569871, 2019.

31. Köhler, S., O. Lieleg, and A.R. Bausch, Rheological characterization of the bundling transition in F-actin solutions induced by methylcellulose. PLoS One, 2008. 3(7): p. e2736.